# CoPeP: Benchmarking Continual Pretraining for Protein Language Models

## Abstract

In recent years, protein language models (pLMs) have gained significant attention for their ability to capture the structure and function of proteins, accelerating the discovery of new therapeutic drugs. These models are typically trained on large, evolving corpora of proteins that are continuously updated by the biology community. The dynamic nature of these datasets motivates the need for continual learning, not only to keep up with the ever-growing dataset sizes, but also as an opportunity to take advantage of the temporal meta-information that is created during this process. As a result, we introduce the Continual Pretraining of Protein Language Models (CoPeP) benchmark, a novel benchmark for evaluating continual learning approaches on pLMs. Specifically, we curate a sequence of protein datasets from the UniProt database spanning 10 years and define metrics to assess the performance of pLMs on diverse protein understanding tasks. We evaluate several methods from the continual learning literature, including replay, unlearning, and plasticity-based methods, some of which have never been applied to models and data of this scale. Our findings reveal that incorporating temporal meta-information improves perplexity by up to 20% over training from scratch on the latest snapshot of the database, and that several continual learning-based methods outperform naive continual pretraining. The CoPeP benchmark presents an exciting opportunity for studying these methods at scale on an impactful, real-world application.

## 1 Introduction

Proteins are the fundamental building blocks of life, acting as the primary machinery of all living organisms. Their function is mostly determined by their three-dimensional shape, which in turn is encoded into a linear sequence of 20 distinct amino acids. Predicting the properties of a protein from its sequence is one of the core challenges in computational biology. Recently, protein language models (pLMs) have emerged as an effective and scalable solution (Rives et al., 2021; Lin et al., 2023; Madani et al., 2023; Nijkamp et al., 2023; Fournier et al., 2024). By treating proteins as a language where amino acids are the "letters", assembling into regions as "words", themselves assembling into whole proteins as "sentences", pLMs can discover the relationship between sequence, structure, and function from large databases (Rives et al., 2021; Notin et al., 2023). This allows them to accurately predict a protein's properties and even to design new proteins for specific applications (Hayes et al., 2025), greatly accelerating drug discovery.

Despite their effectiveness, pLMs face a significant challenge in the dynamic nature of their training data (Fournier et al., 2024). These models are typically trained on enormous, ever-expanding public databases like the UniProt Knowledgebase (The UniProt Consortium, 2025), which are continuously updated. Each year, millions of new protein sequences are deposited, and millions of others are curated out after being identified as non-proteins. Consequently, the practice of retraining models from scratch on each new data release is becoming computationally prohibitive. This challenge, however, also presents a unique opportunity. The temporal evolution of these databases provides valuable metadata. Sequences that persist over time serve as strong examples of true proteins, while those that are later curated out can be treated as implicit examples of likely non-proteins. By leveraging this history, a model can more effectively learn the language of proteins.

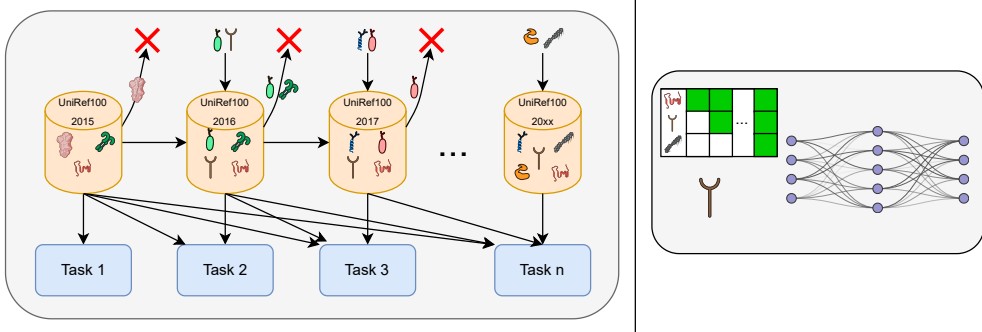

Figure 1: **Left:** The benchmark data process. For every year, we pull the latest UniRef100 release, which reflects the continuous discovery and curation of proteins by biologists. Each task in our benchmark is cumulative, containing all proteins from the start date up to a given year. **Right:** The model training setup. A pLM is trained on all protein sequences and can optionally leverage temporal metadata, such as the number of releases each protein has been a part of, which can be used by methods such as temporally weighted replay, model unlearning.

Although continual learning is a well-established field (Wang et al., 2024) with many artificial benchmarks, there is a growing demand for more realistic alternatives. Indeed, while these controlled environments are perfect for measuring loss of plasticity and catastrophic forgetting, they do not reflect the scale and complexity of real-world data. With the rise of Large Language Models (LLMs), there has been much interest in the community to explore ways to continually update these models with new information. One approach involves limited unlearning or updating a small number of facts that the model might have memorized (Bourtoule et al., 2021; Yao et al., 2023). Other works try to extend the pretraining process itself with new datasets (Gupta et al., 2023; Abbes et al., 2025; Ke et al., 2022; Li et al., 2025). Despite this interest, however, there are not many general-purpose continual pretraining datasets where the goal is to extend the pretraining phase, and most academic works end up using domain-adaptive pretraining setups (Ke et al., 2022; Yıldız et al., 2025).

To bridge this gap, we introduce the Continual Pretraining for Protein Language Models (CoPeP) benchmark. CoPeP provides a realistic, large-scale solution for evaluating continual learning approaches on pLMs. We curate a sequence of protein datasets from 8 yearly releases of the UniProt database, giving us a unique opportunity to study how models adapt to continuously evolving data. We evaluate several state-of-the-art methods from the continual learning literature, including Gradient Ascent (Golatkar et al., 2020), Hare Tortoise (Lee et al., 2024), Replay (Rolnick et al., 2019; Chaudhry et al., 2019), and Shrink and Perturb (Ash & Adams, 2020), applying some of them for the first time at a scale comparable to real-world applications. We evaluate our models on two types of tasks: (1) a high-quality validation set of experimentally verified proteins to assess performance on natural protein distributions (Fournier et al., 2024); (2) ProteinGym (Notin et al., 2023), which benchmarks the ability to predict the effects of protein mutations. Our findings reveal that several of these methods improve performance over naive continual pretraining, and that leveraging temporal metadata yields a measurable improvement over models trained on individual years.

Our contributions are three-fold. First, we introduce CoPeP, a new benchmark for continual learning on real-world protein databases. Second, we are the first to apply and evaluate several state-of-the-art continual learning methods on a problem of this scale and complexity. Finally, we demonstrate that temporal metadata contained in the history of proteins being added or removed from the database can be leveraged to improve the performance of pLMs beyond that of single years.

## 2 RELATED WORK

**Continual Learning and Model Updating** Continual learning is a machine learning paradigm in which models are trained incrementally on a sequence of data or tasks, aiming to accumulate and update knowledge continuously much like humans do. Research in this area primarily focuses on two key challenges: catastrophic forgetting, the loss of previously acquired knowledge (McCloskey

& Cohen, 1989; Kirkpatrick et al., 2017), and loss of plasticity, the reduced ability to adapt to new data (Dohare et al., 2024). While some studies investigate continual learning under natural data shifts (Koh et al., 2021; Lin et al., 2021; Cai et al., 2021; Bornschein et al., 2023), the datasets used are typically much smaller than modern pretraining corpora. Most of the research in continual learning considers smaller academic datasets like CIFAR-10 and MNIST (Goodfellow et al., 2013; Zenke et al., 2017; Krizhevsky et al., 2009; Rebuffi et al., 2017) that allow for controlled experimental setups and the study of severe distribution shifts that may be rare in natural data. However, the limited scale of these datasets raises questions about how well existing methods generalize to larger and more complex scenarios.

More recently, the field has started to shift toward updating large pretrained models. This includes model editing, which updates specific facts in the model without full retraining (Meng et al., 2022; Mitchell et al., 2022), and model unlearning, which aims to remove the influence of specific data points (Bourtoule et al., 2021; Jang et al., 2023). Another line of work involves continually fine-tuning a pretrained model across a sequence of downstream tasks (Jin et al., 2021). Of particular relevance to our work is continual pretraining, where the pretraining process itself is extended to incorporate new data. This has been explored in domain-adaptive pretraining, in which models are sequentially trained on datasets from distinct, specialized domains (Gururangan et al., 2020; Chalkidis et al., 2020). However, these domains are often narrow in scope, and the datasets involved remain relatively small compared to those used in general pretraining. A notable exception is the work of Gupta et al. (2023), which studied the dynamics of training a large model on two datasets in sequence. Nevertheless, practical applications often require methods that scale to much longer sequences of datasets.

**Protein Language Models**   Research in natural language processing (NLP) has recently been adapted to biology by treating the amino-acid sequence of proteins as a form of language. This perspective has led to the development of protein language models (pLMs), biologically inspired analogues of NLP models. For example, the autoregressive ProGen2 (Nijkamp et al., 2023) is based on GPT-2 (Radford et al.), while the masked ESM (Rives et al., 2021; Lin et al., 2023) and AMPLIFY (Fournier et al., 2024) draw inspiration from BERT (Devlin et al., 2019). Trained on large, diverse, and ever-growing protein sequence databases (Suzek et al., 2015; Jumper et al., 2021; Richardson et al., 2023), these models aim to capture evolutionary relationships and discover the underlying principles that govern protein structure and function. This approach has made pLMs an essential tool in computational biology for a wide range of applications such as mutational effect prediction, protein structure modeling, and de novo protein design (Hayes et al., 2025).

To evaluate the capabilities of protein language models, the community relies on several specialized benchmarks targeting different aspects of protein understanding. For protein folding, the Critical Assessment of protein Structure Prediction (CASP) is a biannual challenge that tests a model's ability to predict 3D structures from amino acid sequences (J et al., 2018). For protein engineering and fitness prediction, the ProteinGym benchmark assesses how accurately models can predict the functional effects of mutations (Notin et al., 2023). In addition, broader multi-task benchmarks like TAPE (Rao et al., 2019) and PEER (Xu et al., 2022) evaluate model performance across a wide range of tasks, including remote homology detection and secondary structure prediction. In this work, we focus specifically on protein engineering and fitness prediction, given its crucial role in the drug discovery pipeline.

## 3   CoPeP Benchmark

To bridge the gap between continual learning research and its practical application, we introduce CoPeP, the Continual Pretraining for Protein Language Models benchmark. Built from successive UniProt releases, CoPeP reflects the challenge of keeping models updated with rapidly evolving biological data. It serves as a complex and large-scale testbed for continual learning methods, with significant implications for protein modeling and drug discovery.

### 3.1   Dataset

The CoPeP benchmark is constructed from the UniRef100 database (Suzek et al., 2015), which aggregates and clusters protein sequences curated by the UniProt Knowledgebase (The UniProt

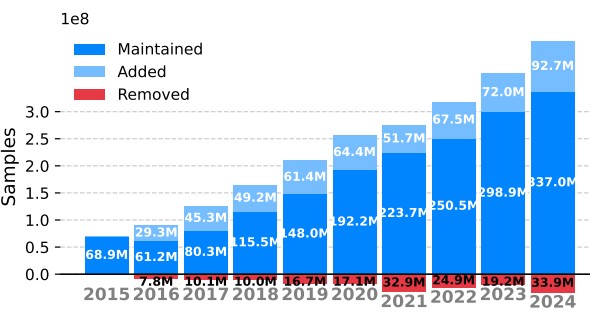

Figure 2: The number of proteins newly selected for and those no longer present in UniRef100 for each year in the benchmark. Despite millions of proteins being removed each year, the size of the dataset still grows as even more proteins are added.

Consortium, 2025), and constitutes the bulk of the training data for several pLMs (Rives et al., 2021; Lin et al., 2022; Fournier et al., 2024; Nijkamp et al., 2023). UniProt is updated multiple times each year, with millions of sequences added, removed, or replaced to reflect new biological knowledge and improved data quality. This evolving nature makes it an ideal foundation for evaluating continual pretraining.

For CoPeP, we select 10 consecutive yearly UniRef100 releases from 2015 to 2024, each corresponding to one task in the benchmark (the specific release and dates are listed in Table 4). These releases span hundreds of millions of protein sequences, with the dataset size increasing substantially year over year (Figure 2). Importantly, proteins may appear, disappear, or persist across releases: new sequences are introduced as biological discoveries accumulate, while others are removed if later deemed redundant or incorrect. Moreover, the dataset size does not grow linearly over time as, each year, an increasing number of new samples is added to the dataset.

Each sample is associated with an identifier and a protein sequence. However, the same identifier can map to multiple sequences, and vice-versa. To ensure consistency, we remove duplicate entries where both identifier and sequence are exact matches. Each dataset thus represents a faithful snapshot of the biological knowledge available at that time, capturing both growth in coverage and changes in curation practices. Together, these sequential datasets define the CoPeP training stream, providing a realistic setting to investigate how continual learning methods cope with evolving large-scale corpora.

## 3.2 STREAMING PROTOCOL

In traditional continual learning setups, training proceeds over a sequence $\mathcal{D}_1, \ldots, \mathcal{D}_n$ of $n$ tasks, where each dataset $\mathcal{D}_i = \{x_j\}_{j=1}^{m_i}$ is drawn from a task-specific data distribution $x \sim \mathcal{P}_i$. The challenge typically arises from distribution shifts between tasks, i.e., $\mathcal{P}_i \neq \mathcal{P}_{i+1}$, which force the model to balance stability (retaining knowledge of earlier tasks) with plasticity (adapting to new tasks).

In CoPeP, the structure is slightly different. We also define a sequence $\mathcal{D}_1, \ldots, \mathcal{D}_n$ of $n$ tasks, where each $\mathcal{D}_i$ corresponds to the UniRef release from year $i$. However, in our case, these datasets are noisy snapshots of a common (unknown) underlying distribution $\mathcal{P}^*$. Importantly, the noise is systematic rather than random, as the protein datasets evolve over in a way reflecting community knowledge and interest. However, it is unknown how representative $\mathcal{D}_i$ is of $\mathcal{P}^*$, with the challenge that yearly increments of the dataset do not correlate with improvements of $\mathcal{P}_i$ w.r.t. $\mathcal{P}^*$ (Fournier et al., 2024; Spinner et al., 2025).

Another difference between our setup and previous continual learning setups is that CoPeP does not forbid access to past data. Rather, at for task $i$, the learner may leverage the union of all observed datasets $\mathcal{U}_i = \bigcup_{j=1}^{i} \mathcal{D}_j$. This makes it possible to exploit temporal meta-information about the samples, such as the *multiplicity* $c(x)$ of a sample, which counts how many consecutive years a protein has persisted in UniRef, i.e., $c(x) = \sum_{i=1}^{k} \mathbb{I}_{\mathcal{D}_i}(x)$. Such information provides a signal

of sequence reliability, distinguishing consistently validated proteins from those that appear only transiently.

By structuring the problem this way, CoPeP reflects the practical challenges of maintaining large-scale models under real-world data evolution, while retaining the core challenges of continual learning paradigms.

### 3.3 EVALUATION

Unlike traditional continual learning setups, because the underlying distribution that we are trying to learn is the same across all tasks, we are not concerned with metrics such as forgetting or transfer. Instead, at each evaluation timestep, we only measure the performance of the model on our suite of evaluation tasks at that specific timestep.

**Validation Set** We use the UniRef validation set introduced in Fournier et al. (2024) as part of our evaluations. These sequences were curated to be high-quality, complete proteins with strong experimental evidence for their existence. We deduplicated all of our training data against this validation set at the 90% sequence identity level using MMSeqs2 (Steinegger & Söding, 2017; Kallenborn et al., 2025) to ensure that the proteins in this validation set are not seen by the models at training time. The UniRef dataset contains proteins from all three domains of the phylogenetic tree of life (Bacteria, Archaea and Eukarya). Thus, performance on this set is an indicator of how well the model is able to reconstruct a broad range of proteins. We track both validation perplexity and accuracy.

**ProteinGym** ProteinGym (Notin et al., 2023) is a broad benchmark designed for protein design and fitness prediction. It contains millions of mutated sequences from 217 deep mutational scanning assays across different taxa (humans, other eukaryotes, prokaryotes, and viruses). For each original sequence, the model ranks the mutations of that sequence by how likely they are, and this ranking is compared to ground truth rankings generated from experimental data and clinical annotations, computing the Spearman's $\rho$ between the two rankings. Across the set of assays, this results in more than 217 Spearman rank coefficients which we aggregate and report.

**PEER** PEER (Xu et al., 2022) is a multi-task benchmark for protein understanding. It contains 17 tasks spanning various aspects of protein function prediction, protein localization prediction, protein structure prediction, protein-protein interaction prediction and protein-ligand interaction prediction. Because our model only operates on protein sequences, we exclude the protein-ligand interaction tasks, and we also exclude the ProteinNet based contact prediction task due to computational constraints. This leaves us with 14 tasks from the original benchmark. Each task has its own evaluation metric, which we follow and report. We also report the average win rate of each model across all tasks compared to the other baselines, i.e. for each year, what percentage of the other models it outperforms.

**DGEB** DGEB (West-Roberts et al., 2024) is another multi-task benchmark designed to evaluate protein language models using diverse sequences across the tree of life and diverse tasks that capture different aspects of biological function. The benchmark contains 18 tasks, of which 16 use the amino acid modality. These tasks include classification, BiGene mining, Evolutionary Distance Similarity (EDS), pair classification, clustering, and retrieval. Similar to PEER, each task has its own evaluation metric, which we follow and report, along with the win rate for each method across all tasks.

### 3.4 BASE EXPERIMENTAL SETUP

The base model for all of our experiments is the AMPLIFY-120M (Fournier et al., 2024). It is an encoder model based on the BERT transformer (Devlin et al., 2019). For each task, we train for 100k steps using the AdamW optimizer (Loshchilov & Hutter, 2018) with weight decay set to .01 and an effective batch size of 4096. Fournier et al. (2024) use a cosine learning rate decay schedule, however, given the difficulty of rewarming up the learning rate after decay in continual learning (Gupta et al., 2023), we opt to use the warmup-stable-decay schedule (Hu et al., 2024; Li et al., 2025) which is more conducive to continuous training. For the first task, we linearly warm up the learning rate in the first 10k steps to .0005. At the end of each task, we linearly decay the learning rate to 0 over the final 10k steps of the task. When restarting training for the next task, we reset to the pre-decay checkpoint (i.e. the checkpoint right before the learning rate decay at 90k

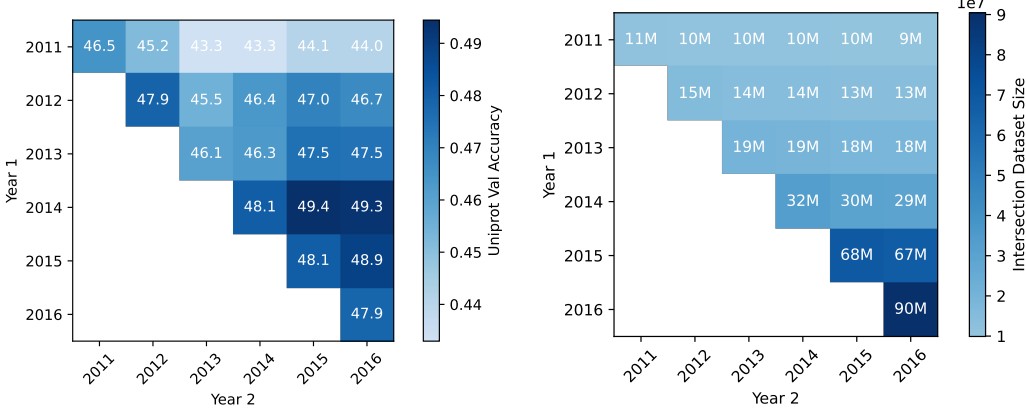

(a) Validation accuracy on the UniProt validation dataset for the filtered datasets. The diagonals are unfiltered yearly releases, while each square in the top half shows the accuracy when the model was trained on the intersection of the data in the two years.

(b) Dataset sizes for filtered data experiments. Each square shows the number of protein sequences used in the training of the models in Figure 3a.

Figure 3: We train models on datasets that are the intersection of two yearly releases. Despite this filtering process creating smaller datasets, the validation accuracy actually improves for most years.

steps into training on the task). Thus, when starting on the sixth task in the benchmark, even though we have done 600k steps of training, the checkpoint we start with has only done 540k gradient steps.

## 3.5 USING TEMPORAL META-INFORMATION

As a preliminary experiment to validate the usefulness of the temporal meta-information described in Section 3.1, we test the hypothesis that data that stays in the UniProt database for longer is of higher quality and leads to better models. For this study, we take the releases for the first two years of our benchmark and releases from the four years prior to our benchmark (i.e. 2011-2016). For each pair of years, we only train on protein sequences that are in the intersection of both releases. Each model is trained for 100k steps according to the procedure outlined in Section 3.4. There are two hypothetical competing effects that this filtering could have: the smaller dataset size could lead to a decrease in performance or the potential increase in quality of data could lead to an increase in performance. We see the results in Figure 3. The performance of models trained on the unfiltered versions of the dataset are along the diagonal. From 2013 onwards, there is an increase in performance going from the unfiltered to filtered version of the datasets, even though the filtered datasets are smaller, implying that the benefit of the higher data quality wins out. For the first two years, since the datasets are already fairly small to begin with, filtering to an even smaller dataset has a deleterious effect. Across all years, however, we see there is an eventual increase in performance as you filter across a longer timespan. Furthermore, the best performance across all datasets comes from the intersection of 2014 and 2015 data, even though that dataset is a fraction of the size of the 2015 dataset, clearly showing the value of using temporal meta-information about the proteins.

## 4 METHODS

As shown in Section 3.5, the curation of data through the yearly updates of the UniProt Knowledgebase affects the prediction accuracy of the trained model. We now perform a large-scale study of 6 different methods to continually pretrain the AMPLIFY-120M model, that takes into account this temporality information. We focus on a set of representative methods spanning across 3 groups: continual learning, plasticity-focused and unlearning methods, with 2 algorithms for each group. Finally, we compare these methods with individual models trained on each yearly release separately, as is current standard practice (Fournier et al., 2024; Hayes et al., 2025) and with a model trained jointly on all data from 2015-2024.

### 4.1 CONTINUAL LEARNING

**Sequential Training** This method is the simple baseline of training on each dataset in sequence, without any additional interventions or regularization. There are no additional hyperparameters for this method.

**Temporally Weighted Replay** Experience Replay (Rolnick et al., 2019; Abbes et al., 2025) is a commonly used technique in continual learning where a small subset of data from previous tasks is saved and rehearsed by the model while training on future tasks to prevent catastrophic forgetting. Given we can access all previous datasets and based on the results in Section 3.5, we use a modified version of this idea where we do not limit ourselves to a fixed size replay buffer. Instead, we continue sampling all samples according to how many previous datasets they appeared in. Let $S = \{\mathcal{D}_1, \mathcal{D}_2, \ldots, \mathcal{D}_{t-1}\}$ be the sequence of datasets up until the current task, and let $U = \bigcup_{i=1}^{t-1} \mathcal{D}_i$ be their union. For any example $x \in U$, let its multiplicity be $c(x) = \sum_{i=1}^{k} \mathbb{I}_{\mathcal{D}_i}(x)$, where $\mathbb{I}_{\mathcal{D}_i}(x)$ is the indicator function. The probability of sampling an example $x$ from $U$ is proportional to its multiplicity and is given by: $P(x) = \frac{c(x)}{\sum_{y \in U} c(y)} = \frac{\sum_{i=1}^{k} \mathbb{I}_{D_i}(x)}{\sum_{i=1}^{k} |D_i|}$. The total loss is given by $(1 - \lambda_{replay})\mathcal{L}_{ce}(b_i) + \lambda_{replay}\mathcal{L}_{ce}(b_{replay})$, where $b_i$ is a batch sampled from the novel protein sequences added in task $i$, $b_{replay}$ a batch of samples from previous tasks sampled according to their multiplicity, and $\lambda_{replay}$ weights the importance of current task compared to previous tasks.

### 4.2 PLASTICITY

Loss of plasticity is a phenomenon in continual learning where as the model trains, it becomes less able to adapt to changes in data distributions. The plasticity preserving methods we use in our experiments are agnostic to the past data distributions and do not use any extra data.

**Shrink and Perturb** Shrink and Perturb (Ash & Adams, 2020) involves periodically shrinking and then adding noise to the weights of a neural network as a means of restoring plasticity to the network. In our experiments, at the start of every task, we set the weights as $\theta_t = \lambda_{shrink}\theta_{t-1} + \lambda_{noise}p$, where $p$ are random weights drawn from the initialization distribution of the network.

**Hare and Tortoise** Hare and Tortoise (Lee et al., 2024) maintains two sets of network weights, slow and fast. The slow weights are an exponential moving average of the fast weights, i.e. at every step the slow weights are set to $\theta_{slow} = \lambda_{ht\_mom}\theta_{slow} + (1 - \lambda_{ht\_mom})\theta_{fast}$. Periodically, the fast weights are reset to the slow weights according to $\lambda_{reset\_freq}$.

### 4.3 UNLEARNING

Unlearning involves actively trying to remove knowledge about specific samples from the network. In our experiments, the forget set for task $t$, $\mathcal{F}_t$ is defined as the set of examples present in task $t-1$ but not in task $t$. With each step, we sample one batch from the current task $b_i \sim \mathcal{D}_t$ and one batch from the forget set $b_{forget} \sim \mathcal{F}_t$.

**Gradient Ascent** Gradient ascent (Golatkar et al., 2020) attempts to unlearn knowledge by performing a gradient ascent step on data from the forget set. To prevent divergence, it also performs a descent step on data that is to be retained. This is implemented as optimizing the following loss: $\mathcal{L}_{ce}(b_i) - \lambda_{asc}\mathcal{L}_{ce}(b_{forget})$ where $\mathcal{L}_{ce}$ is the standard cross entropy loss used in training.

**Random Labels** Random labels (Golatkar et al., 2020) tries removing the knowledge in the forget set by sampling the targets of the forget set from the uniform distribution and performing gradient steps. The loss for the forget set is weighted by $\lambda_{rand}$.

### 4.4 DESCRIPTION OF HYPERPARAMETER SEARCH AND OTHER EXPERIMENTAL DETAILS

For each method, we use the same base hyperparameters (e.g. learning rate, weight decay, batch size), and search over the method specific hyperparameters. Given the fact that several of these methods have not been used on such a scale, there does not exist much guidance in the literature on suitable ranges for many of these hyperparameters. We instead use an iterative, pruning based approach to our hyperparameter search to try a wide range for each method, and quickly prune suboptimal configurations. For each method, we evaluate 8 random configurations at 50k steps of

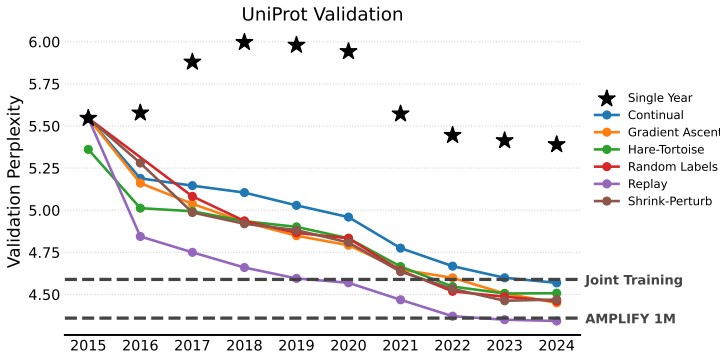

Figure 4: Validation perplexity on the UniProt validation set described in Section 3.3. All continual methods in our study beat the naive continual baseline and single year baseline.

total training. We then seed a Bayesian sampler with the results of those trials and sample 8 more configurations that are also evaluated at 50k steps of total training. The best 4 configurations from the 16 total trials are trained for another 150k steps, at which the best configuration is selected and trained for the remaining tasks in the benchmark. No method other than Hare and Tortoise deviates from the standard training for the first task in the benchmark, so every method (except for Hare and Tortoise) is started on task 2 from the pre-decay checkpoint of task 1. Hare and Tortoise is started from scratch on task 1. We use validation loss as the selection criterion in the search.

## 5 RESULTS

### 5.1 UNIPROT VALIDATION SET

In Figure 4, we show the performance of our models on the UniProt validation set. We notice several trends with our results. First, we see that performance generally seems to improve over time for the continual baselines. While this may seem trivial, it validates taking a continual approach to the problem. The steady improvement shows that continual training does not saturate the network or prime the network too heavily so that it cannot learn from future data. Furthermore, there is a big gap between the single year baseline and the continual baselines. This is partly because the continual models trained for longer, but this establishes that training from a continual checkpoint is effective. With each release, if the choice is to train for a certain number of steps from scratch or from a continual checkpoint, the continual checkpoint is a much more effective starting point. Surprisingly, every continual method also outperforms the joint training baseline that was trained on all data from 2015-2024 for the same number of steps as the continual models. This is likely because the joint training baseline also learns from data that was removed, while the continual models only learn from data is present in the current release.

We should also note that several models start reaching the performance level of AMPLIFY 1M (the base model trained for 1 million steps according to Fournier et al. (2024)) with considerably fewer steps and access to less data throughout training. In fact, the temporal replay method essentially matches the performance of AMPLIFY-1M at 8 tasks, which is the equivalent of 730k steps, with much of the training taking place with access to much less data, and overtakes it after the 9th task.

Finally, comparing the methods amongst each other, we see that every method offers better performance compared to the naive continual baseline and (other than the temporal replay baseline) relatively similar performance to each other. This is highly encouraging, as essentially none of these methods were developed for this specific setup, and yet they are all showing positive performance. Hare and Tortoise and Shrink and Perturb are both plasticity preserving methods, but to our knowledge have never been applied to a model or training scale of this size. Gradient Ascent and Random Labels have been used with LLMs, but generally on more limited forget sets and not as a part of continual pretraining. The relative success of these methods shows that all of these approaches to continual learning (forgetting, plasticity, unlearning) have ideas to contribute in this setup.

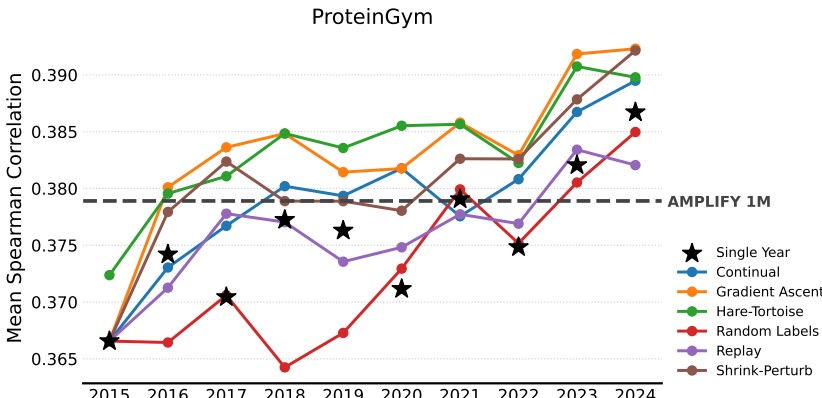

Figure 5: Results on the ProteinGym benchmark. Several continual methods outperform the naive continual baseline, as well as the AMPLIFY-1M baseline, a version of our model that was trained for 1 million steps.

## 5.2 PROTEINGYM

The trends for the ProteinGym evaluation (Figure 5) are slightly harder to define compared to the UniProt Validation set. Gradient Ascent and Shrink and Perturb both perform the best across all methods. All the continual and the single year baseline at 2024 outperforms AMPLIFY-1M, with 3 of the non-naive continual learning methods also outperforming naive continual learning. Surprisingly, the temporal replay method, which does the best on the UniProt validation set, does performs the worst on ProteinGym. We discuss the potential reasons for this in Section 5.4. There does seem to be a measure of early saturation for a while, as the improvement in performance for most of the methods seems to plateau after the first 3 tasks, but then we see a jump in performance at the end when the 2023 and 2024 data is introduced. This is likely because ProteinGym was introduced in 2023, and thus many of the proteins in the benchmark were likely added to the UniProt database in the last two years.

## 5.3 MULTITASK PROTEIN UNDERSTANDING BENCHMARKS

We show the a summary of the results on the multitask protein understanding benchmarks DGEB and PEER in Figures 6a and 6b respectively. The win rates are computed compared to every other checkpoint for all methods and years, across all tasks, and thus is a relatively stable metric. For more fine-grained results on these benchmarks, please see Appendices C and D.

For PEER 6a, none of the Single Year baselines perform significantly better than all the rest, implying that there is not necessarily a year that is better for downstream performance. Some Single Year results are worse than average, however, and it is notable that after training on 2022 data (one of the worst Single Year performances), the performance of the Gradient Ascent method essentially collapsed. This is also seen in the DGEB results 6b. The best performing model at any point was Shrink and Perturb at years 2020-2021, followed by Temporal Replay at year 2023.

For DGEB 6b, we see that the naive continual baseline does not perform well, with nearly all continual learning methods outperforming it (except for Gradient Ascent after training on 2022 data). Interestingly, the best model again is at year 2020, although in this case it is Random Labels that achieves the best performance. Temporal Replay again achieves strong performance at year 2023. Overall, the results for DGEB generally align with the PEER results, but both of these benchmarks seem to be not very aligned with the ProteinGym results.

## 5.4 TRADEOFFS

Drug discovery is a long process and requires many different capabilities with respect to proteins, including generation, property prediction, fitness prediction, and optimization. It is difficult to create

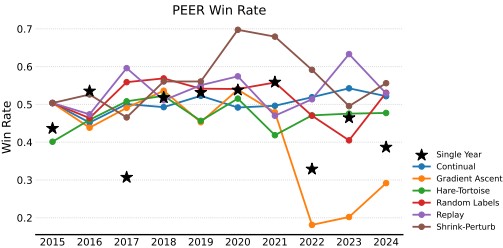 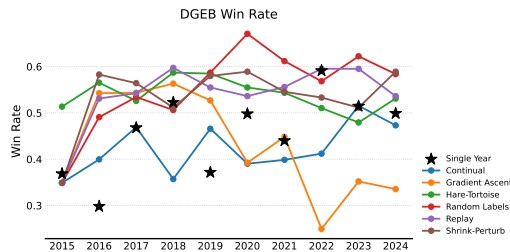

(a) Relative win rates of each method on the PEER benchmark. The best performing win rate is achieved by Shrink and Perturb followed by Temporal Replay.

(b) Relative win rates of each method on the DGEB benchmark. The best performing win rate is achieved by Random Labels. Furthermore, most continual methods match or outperform the naive continual baselines.

Figure 6: Results on the multitask protein understanding benchmarks.

an evaluation that is able to cover all of these capabilities. In this section we discuss the tradeoffs of the two evaluations we use in our benchmark.

Both the UniProt validation set and ProteinGym were curated from natural proteins, which means that models that do well on them would be helpful in creating therapeutic drugs, but not necessarily industrial proteins. The UniProt validation set was constructed to be as close as possible to the natural distribution of proteins, with the idea that nature is a good inductive bias. The proteins were selected to be from diverse and highly studied proteomes. A consequence of the latter point is that the highly studied proteomes were likely in the earlier UniRef100 releases which could explain the success of a method like temporal replay that upweights such samples. Because we deduplicated our training set against the validation set, however, the evaluation rewards models that do not overfit to specific samples or mutations that they see in the data, and instead learn to generalize to the larger patterns. On the other hand, ProteinGym rewards models that can properly evaluate the fitness of specific mutations in a protein. Given we did not deduplicate our training set against ProteinGym and that memorizing specific sequences could provide an advantage to the model, it is also possible that ProteinGym would reward models that overfit the data slightly.

## 6 DISCUSSION

We present CoPeP, a benchmark for continual pretraining of protein language models. The datasets used in our benchmark are curated from the regular releases of UniProt, and thus naturally evolve as the biologist community's knowledge and interest evolve. CoPeP is regularly extensible as each new UniProt release becomes available, making it more difficult to saturate the benchmark. In our work, we show that several different approaches to continual learning and unlearning are able to improve on naive continual learning, and our benchmark is an opportunity for those communities to develop and test their methods on a realistic, large scale setting. Several of the methods we present are also fairly orthogonal to each other, and future work can investigate how to combine them to create a better method. Although not explored in our work, the closely related field of model editing could also potentially apply contribute to this problem.

Our work also explores the idea of using temporal meta-information about each sample to guide training. We use this information as both a filter and as a replay strategy, and and show that both approaches improves performance. Future work should explore protein specific learning methods that can better leverage this temporal meta-information.

We hope that this benchmark can accelerate progress in protein language model learning. For large biomedical companies, it may be cost-feasible to simply retrain from scratch on large data, but having to do so takes time that can lengthen experiment cycles. Effective continual training could also enable academic labs to perform relevant and cost-effective research and further push the frontier of drug discovery.

## 7   REPRODUCIBILITY STATEMENT

The details of our model training and hyperparameter selection are provided in Sections 3.4, 4.4, F.1, and G. The details of our dataset curation are provided in Sections 3.1, 3.3, and H. Upon acceptance, we also intend to release the code, checkpoints, and datasets used to conduct all of our experiments.

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

## A  EVOLUTION OF DATA

### A.1  EMBEDDING VISUALIZATION

In this experiment, we analyze how protein sequence datasets evolve over years by visualizing their structure in embedding space. We use representations from AMPLIFY (trained with 1 million steps) and apply UMAP (McInnes et al., 2018) to project high-dimensional protein embeddings into two dimensions. This enables us to observe broad patterns in the data and how they change across consecutive UniRef100 releases.

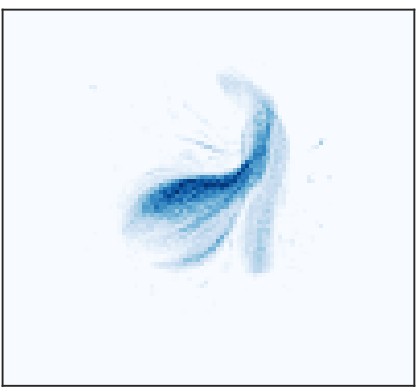

Figure 7: UMAP projection of protein embeddings from all UniRef100 releases (AMPLIFY 1M representations). The plot shows a stable global structure with a dense core and branches, indicating natural groupings of proteins.

Figure 7 shows the embedding of the full dataset i.e., sequences from all years. The plot reveals a global structure with a dense central region and branches, suggesting natural groupings of proteins. Differences in density highlight areas where certain types of sequences are more common.

Each UniRef100 release both adds and removes sequences, reflecting the expansion of biological knowledge and ongoing curation. To illustrate these dynamics, Figure 8 compares additions (blue, top row) with removals (red, bottom row) per year. Overall, while the global structure of protein embeddings is stable, Figure 8 indicates local shifts such as density increases and cluster expansion. This underscores why continual learning is critical for protein language models. Instead of treating each release as an isolated datasets, continual methods can exploit temporal information to adapt to new proteins as well as retain knowledge.

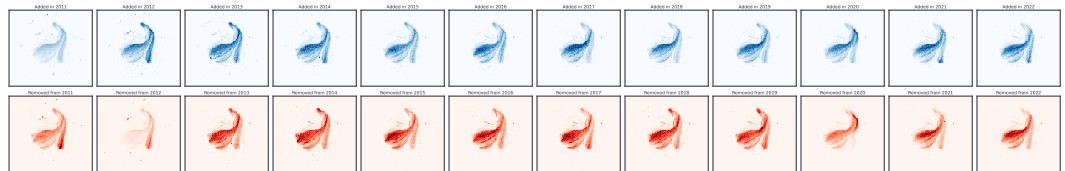

Figure 8: Yearly dynamics of UniRef100 embeddings. Top row (blue): proteins added in each year; bottom row (red): proteins removed. While the global organization of protein embeddings is stable, the local shifts such as density increases and cluster expansion are indicate yearly shift in underlying distribution.

### A.2  MODEL EMBEDDING SHIFTS

In Figure 9, we visualize how different continual learning methods structure their protein embeddings. For each method, we select the final checkpoint, and extract embeddings for a representative subset of proteins that were added and removed across all years. We then apply UMAP to project these embeddings into two dimensions. The color shading indicates the density of proteins from

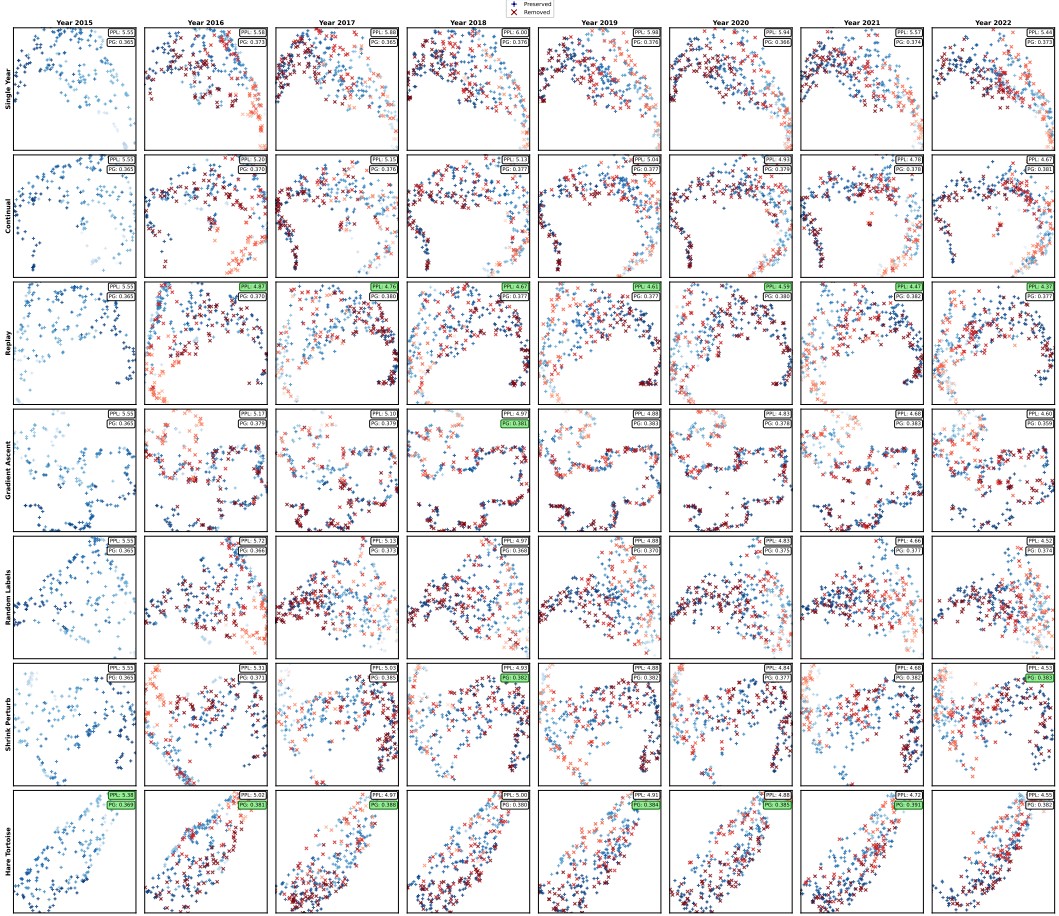

Figure 9: UMAP visualization of protein embeddings across continual learning methods (2015–2022) with performance metrics (PPL: validation perplexity; PG: ProteinGym mean Spearman correlation). Colors shades are based on single year clusters. Darker shades indicate denser parts. Same color shades are used in other rows/methods to indicate those centroid/representative proteins locations.

the single year clusters, with darker shades representing denser regions. Interestingly, most methods maintain a similar global structure except for Gradient Ascent, which shows a more rope-like structure.

## A.3 DATA STATISTICS

In this section, we analyze how various protein sequence statistics evolve over years. We compute several statistics for each protein sequence in the UniRef100 releases from 2015 to 2024 using the Biopython library (Cock et al., 2009). Specifically, we create two sample QQ-plots comparing the distribution of each statistic between each year and the reference year 2015. The statistics we analyze include:

- Aromaticity
- Charge at pH 7
- Instability Index

- Isoelectric Point
- Molar Extinction Coefficient (oxidized and reduced)

- Protein Length
- Longest Repeat Ratio

Overall, we see that most statistics show relatively stable distributions across years, for at least some of the statistics (molar extinction coefficient, protein length, charge at pH 7) there are outliers at

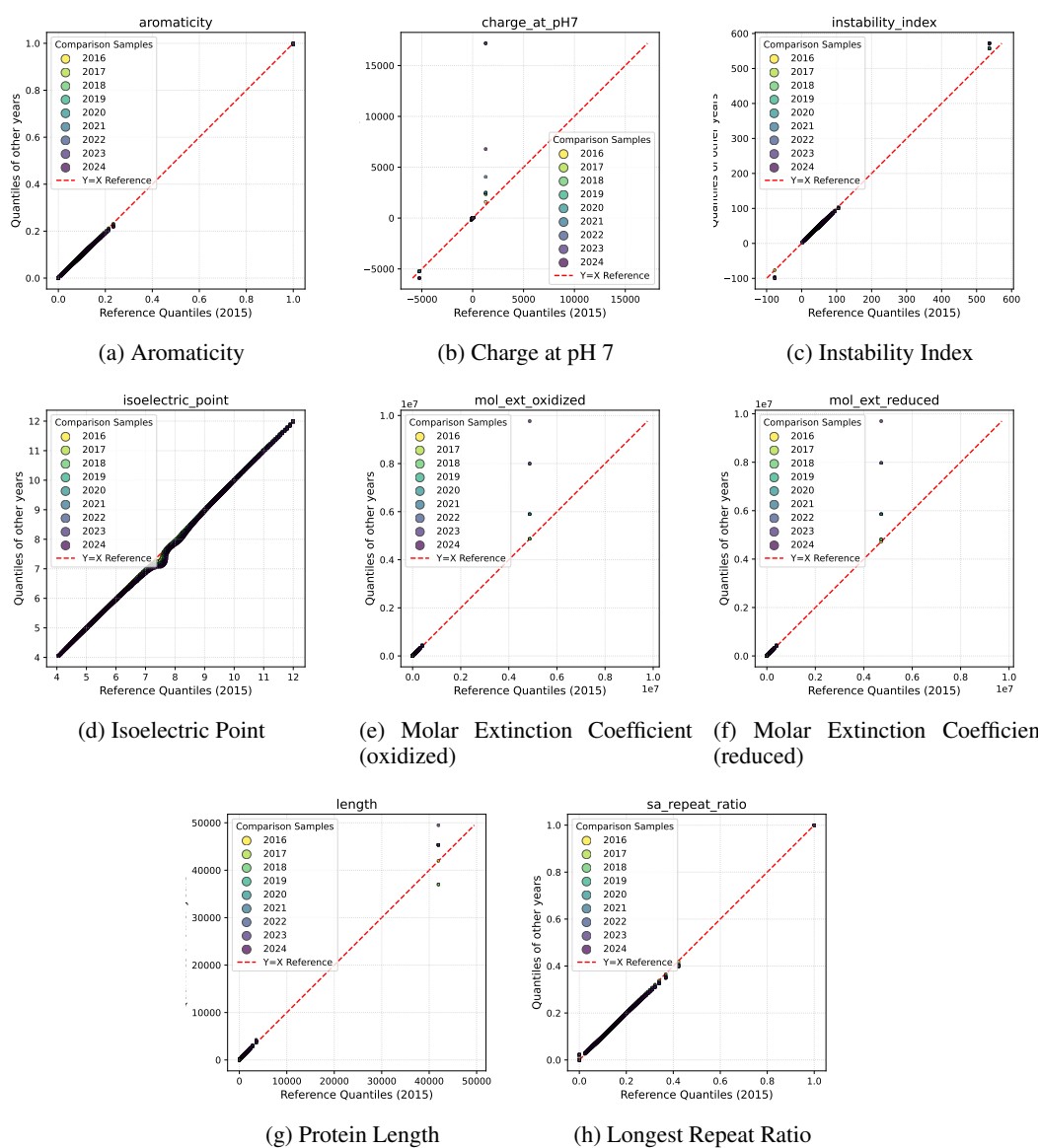

Figure 10: Two sample QQ-plots of various protein sequence statistics across different years. Each line represents a comparison between a year and the reference year 2015.

the extremes of the distributions for several years. Furthermore, there is a noticeable shift in the distribution of the isoelectric point over the years, with later years showing a higher density of proteins with isoelectric points around 7-8.

## B  FINE GRAINED RESULTS ON PROTEINGYM

We also visualized two variants of the ProteinGym evaluation, similar to Figure 5: the best performance achieved in each year of training and the fine-grained trajectory of performance across all steps.

In Figure 11, we observe that Hare Tortoise consistently delivers the strongest results, with Gradient Ascent and Shrink Perturb close behind. All three methods perform better than the AMPLIFY 1M baseline across nearly all year releases, while continual learning and replay result in modest gains.

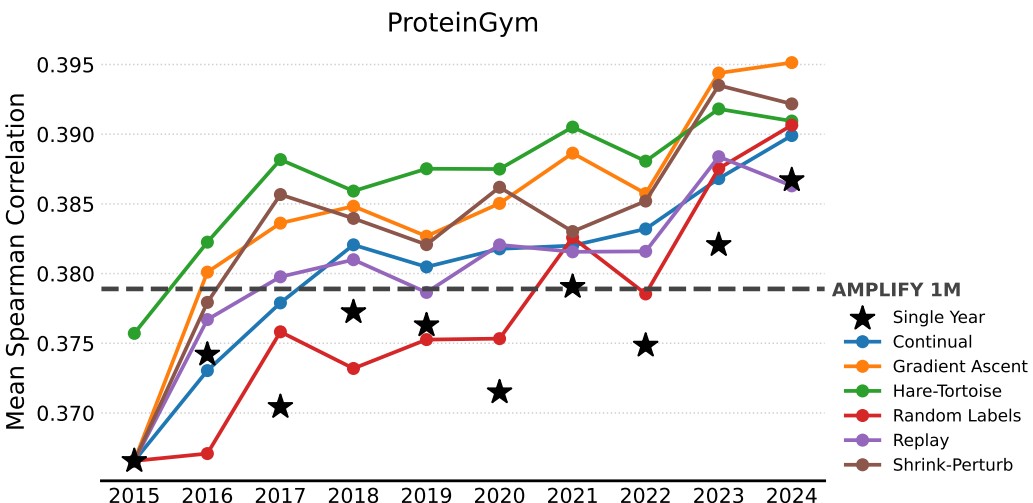

Figure 11: Best mean Spearman correlation for continual training on ProteinGym. Hare Tortoise achieves the best performance across nearly all years, with Gradient Ascent and Shrink Perturb close behind. These methods consistently perform better than AMPLIFY 1M.

Random Labels again shows improvement relative to the naive Single Year baseline, but it does not reach the same level as the other methods.

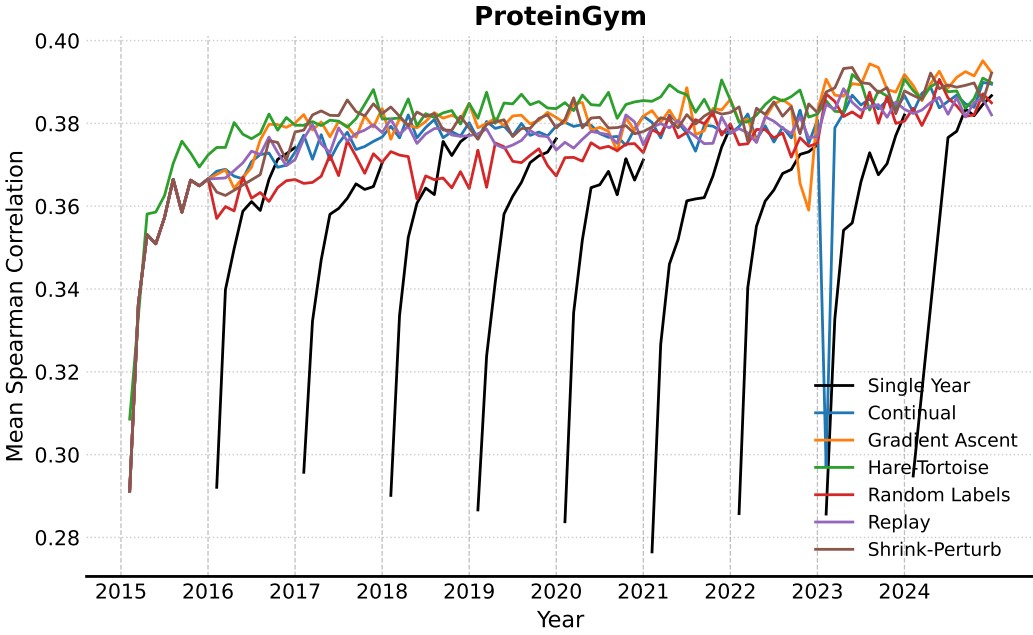

Figure 12: Mean Spearman correlation on ProteinGym across training steps for continual training on ProteinGym. While the naive Single Year baseline resets each year and lags substantially, continual learning methods such as Hare Tortoise, Gradient Ascent, and Shrink Perturb maintain strong performance throughout training and consistently perform better than AMPLIFY 1M.

Figure 12 highlight the shortcomings of Single Year most clearly as the model start from scratch. By contrast, Hare Tortoise, Gradient Ascent, and Shrink Perturb maintain strong performance throughout training, suggesting that these methods provide more stable and reliable learning dynamics.

Apart from these results, we also provide the boxplots of Spearman correlations across methods in Figure 13. In all cases, Hare Tortoise, Gradient Ascent, and Shrink Perturb consistently cluster above the AMPLIFY 1M baseline, with relatively tight distributions indicating robust improvements. Replay and Continual show more variance, often overlapping with the baseline but generally outperforming the naive Single Year approach.

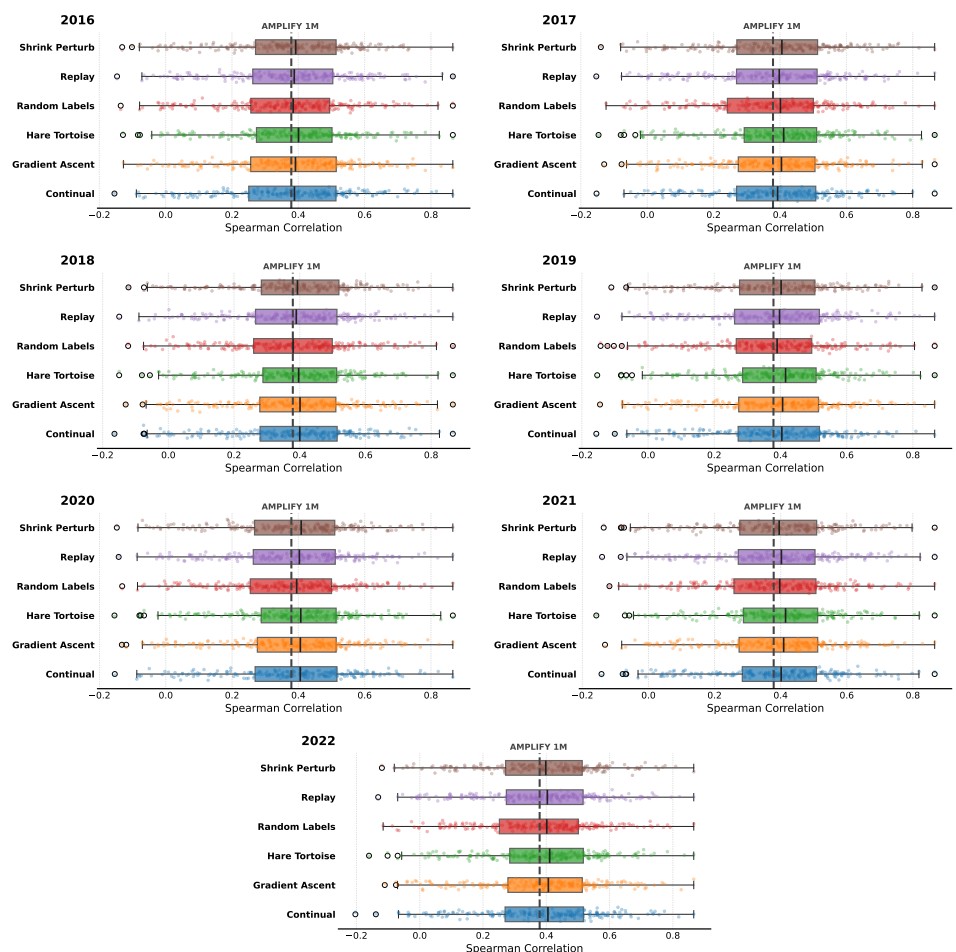

Figure 13: Distribution of Spearman correlations for each method from 2016–2022 on ProteinGym benchmarks. Hare Tortoise, Gradient Ascent, and Shrink Perturb consistently center above the AMPLIFY 1M baseline with tight variance, indicating strong and stable performance.

Table 1 summarizes the AUC performance of different methods on ProteinGym. We again observe that Hare Tortoise consistently achieves the best or tied-best results across nearly all years, with Gradient Ascent and Shrink Perturb closely following. These findings align with the our observations in Figure 5, By contrast, continual learning and replay provide moderate gains over the naive Single Year baseline.

## C    FINE GRAINED RESULTS ON PEER

In Figure 14, we show the full results for each task on the PEER benchmark.

## D    FINE GRAINED RESULTS ON DGEB

In Figure 15, we show the full results for each task on the DGEB

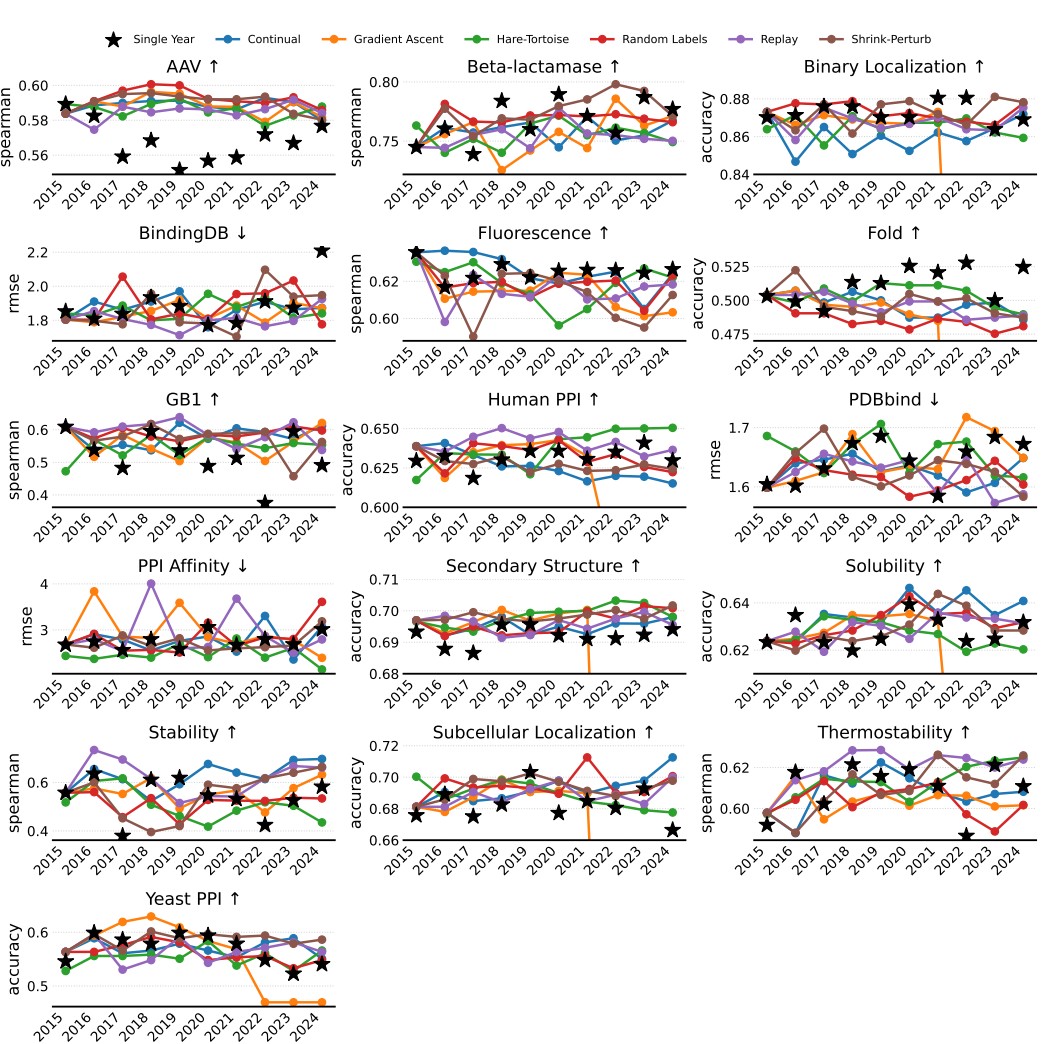

Figure 14: Full results on the PEER benchmark.

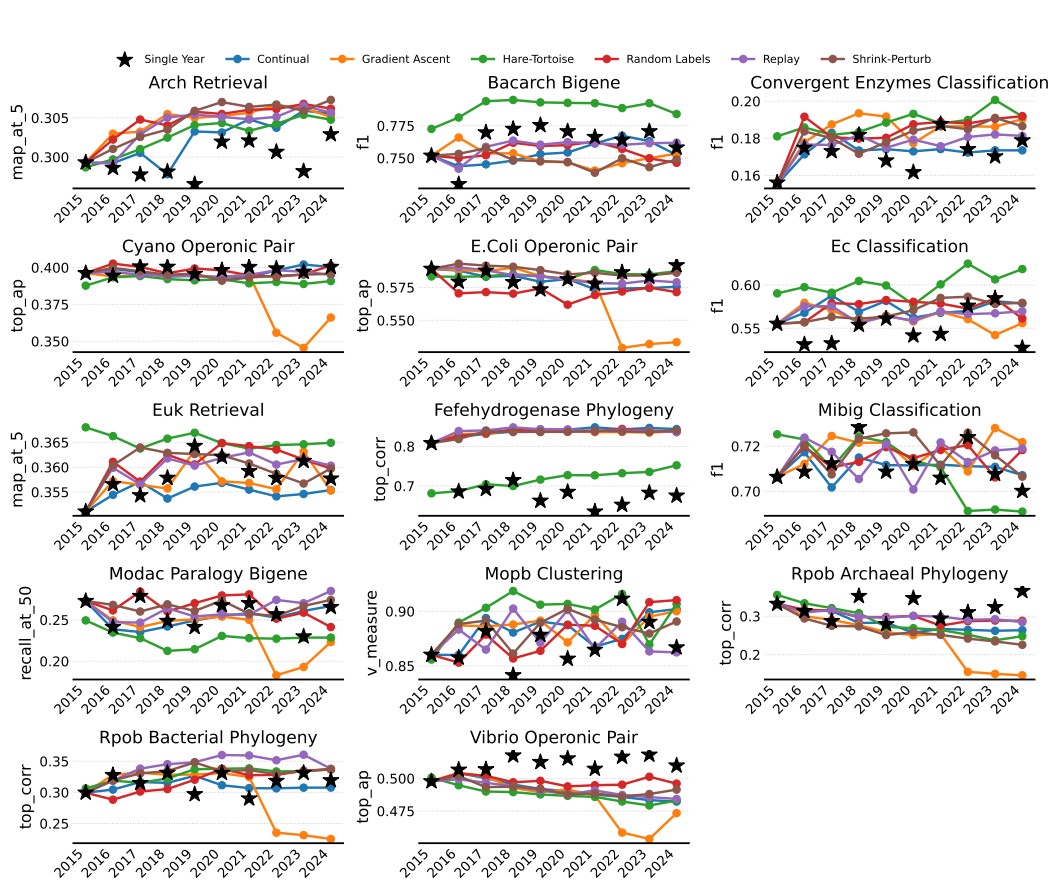

Figure 15: Full results on the DGEB benchmark.

| | Single Year | Continual | Gradient Ascent | Hare Tortoise | Random Labels | Replay | Shrink Perturb |
|---|---|---|---|---|---|---|---|
| 2015 | 0.700 | 0.700 | 0.700 | 0.704 | 0.700 | 0.700 | 0.700 |
| 2016 | 0.705 | 0.703 | 0.707 | 0.708 | 0.701 | 0.704 | 0.707 |
| 2017 | 0.702 | 0.706 | 0.709 | 0.709 | 0.703 | 0.707 | 0.709 |
| 2018 | 0.707 | 0.709 | 0.711 | 0.711 | 0.700 | 0.706 | 0.708 |
| 2019 | 0.707 | 0.708 | 0.708 | 0.710 | 0.701 | 0.704 | 0.708 |
| 2020 | 0.703 | 0.709 | 0.709 | 0.710 | 0.704 | 0.705 | 0.707 |
| 2021 | 0.708 | 0.707 | 0.711 | 0.711 | 0.708 | 0.707 | 0.710 |
| 2022 | 0.705 | 0.709 | 0.710 | 0.709 | 0.705 | 0.706 | 0.709 |

Table 1: Area Under the Curve (AUC) performance on ProteinGym across different methods per year. Consistent with the Spearman correlation in Figure 5, Hare Tortoise achieves the strongest performance across all years, with Gradient Ascent and Shrink Perturb close behind.

# E  UNIPROT VALIDATION RESULTS

## E.1  UNIPROT VALIDATION SET COMPOSITION

In this section, we provide additional information on the makeup of the validation set used Figure 4.

First, we show the taxonomic lineage breakdown of the validation set in Figure 16. We can see that the majority of sequences are Eukaryota, with some coverage of Bacteria and Archaea as well. The most common species in the validation set is Homo sapiens (human), followed by Mus musculus (house mouse) which makes sense given those are likely the most relevant species to drug discovery.

In Figure 17, we look at the pairwise sequence similarity between the different proteins in the validation set, and see that it is quite diverse, with the majority of sequence pairs being around 20-40% similar i.e. in the "protein twilight zone". The "protein twilight zone" refers to the range of low sequence identity (typically 20–35%) where it becomes difficult to determine if two proteins are truly related based on their sequences alone.

## E.2  STRATIFICATION BY LINEAGE

In this section, we present the results on the UniProt validation set for each method stratified by the different lineages present in the dataset. We can see the results in Figure 18. Notably, the mean perplexity tends to follow the perplexity on Eukaryota and Archaea quite well, but Bacteria tends to have a much lower perplexity. Furthermore, as we go down the taxonomic tree, it does not seem to be the case that the model performs significantly better on the more common groups. There is a decent amount of spread in perplexity amongst the common groups, indicating that the models are not just memorizing the most common sequences.

# F  FURTHER ABLATIONS

## F.1  WSD VS COSINE LEARNING RATE

In this section, we clarify the learning rate schedule used by our models. Our model is based off of AMPLIFY (Fournier et al., 2024), which used a cosine learning rate schedule in its training run. Unfortunately, because the cosine learning rate schedule has a fixed span it is unsuitable for continual training. Instead we use the warmup-stable-decay (WSD) schedule which has been used for continual pretraining (Li et al., 2025). In Figure 19, we can see that after decay, the two schedules perform about equivalently.

In our experiments, after each decay period, we reset to the checkpoint right before the decay before moving to the next task. Thus, only 90k out of the 100k gradient steps on a task are used to contribute to the continual training, but it offers a good balance between needing to decay the learning rate and being able to restart the run.

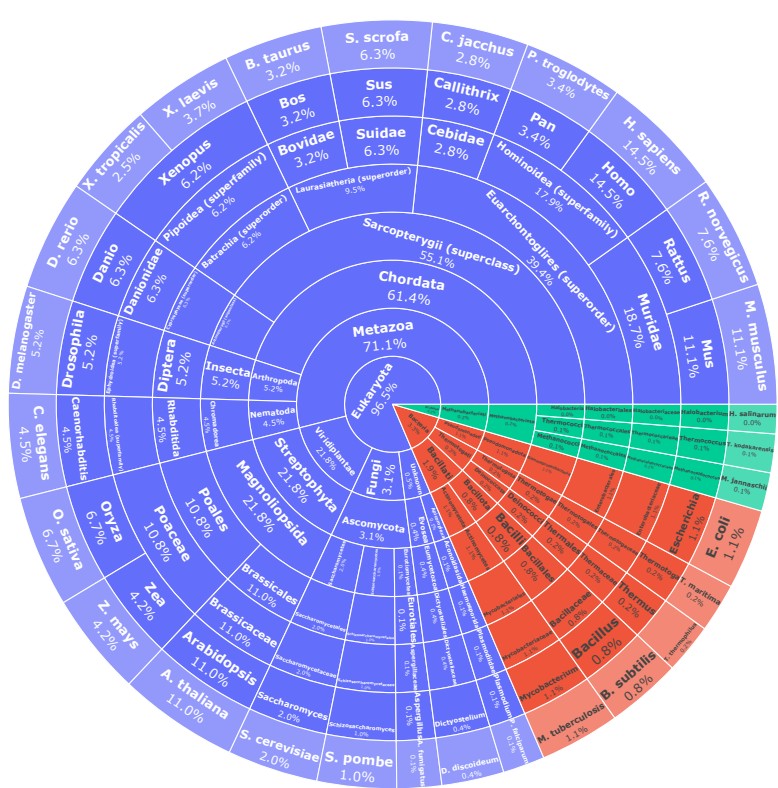

Figure 16: Proportion of different lineages in the UniProt validation set. Note in order to make Archaea properly visible, the area for each sector is according to the log of the number of sequences in that lineage.

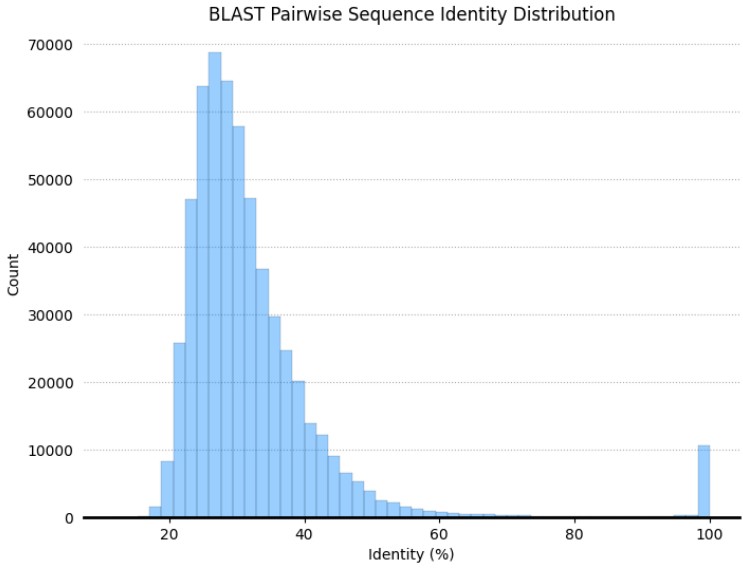

Figure 17: Sequence similarity between different proteins in the validation set.

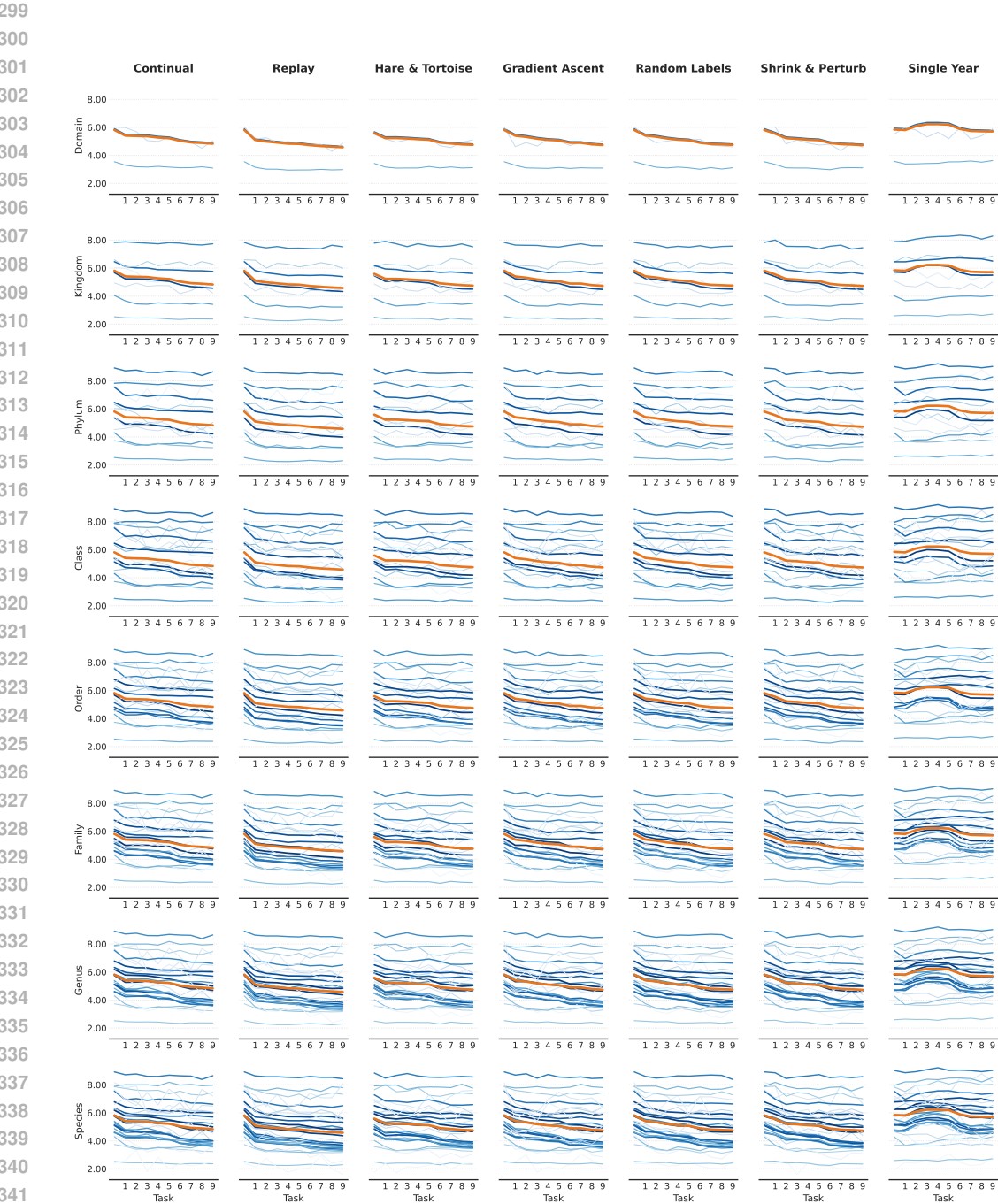

Figure 18: Perplexity on the UniProt validation set broken down by taxonomic lineage for each method. In each subplot, the mean perplexity is shown in orange, and the the lines for the more common groups are shaded to be darker and thicker.

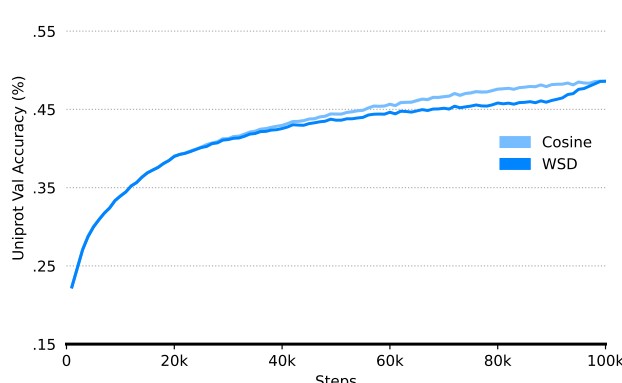

Figure 19: Both the cosine learning rate schedule and the warmup stable decay achieve approximately the same performance.

| Method | UniProt Perplexity |
|---|---|
| 2015 Data (910k steps) | 4.535 |
| Continual Sequence (910k steps) | 4.568 |
| Replay (910k steps) | **4.342** |
| AMPLIFY-1M (1 million steps) | 4.359 |
| Gradient Ascent (910k steps) | 4.450 |
| Hare and Tortoise (910k steps) | 4.507 |
| Random Labels (910k steps) | 4.460 |
| Shrink and Perturb (910k steps) | 4.470 |

Table 2: Results of training for an equivalent number of on a single year (2015) compared to continual training across all years.

## F.2 LONGER TRAINING OF A SINGLE YEAR

In Table 2, we compare training for a longer period on a single year (2015) to continual training across all years. We find that training longer on a single year matches performance of continual training. This might be because the 2015 data is particularly representative of the overall UniProt distribution, as seen in Figure 3a. Regardless, all of the continual learning methods outperform the longer single year training, indicating that continual learning is beneficial beyond just training for longer.

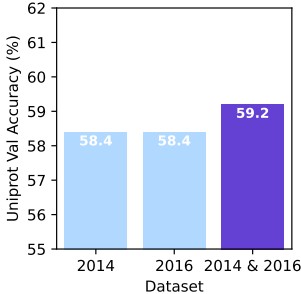

Figure 20: Data filtering experiments with a larger model (350M parameters).

| Method | Hyperparameter | Distribution | Selected Value |
|---|---|---|---|
| Continual | None | | |
| Replay | $\lambda_{replay}$ | Uniform(0, 1.0) | 0.357495045651384 |
| Hare and Tortoise | $\lambda_{ht\_mom}$ | Uniform(.5, 1.0) | .931247906596137 |
| | $\lambda_{reset\_freq}$ | LogInt(10, 10000) | 559 |
| Gradient Ascent | $\lambda_{asc}$ | Uniform(0, 1.0) | 0.0150798214665966 |
| Random Labels | $\lambda_{rand}$ | Uniform(0, 1.0) | 0.00176366392582128 |
| Shrink and Perturb | $\lambda_{shrink}$ | Uniform(0, 0.9) | 0.310430229773085 |
| | $\lambda_{noise}$ | Uniform(0, 1.0) | 0.713412708958246 |

Table 3: The hyperparameter ranges and the selected hyperparameters for each method in our study.

| Year | Release | Date | Number of Proteins |
|---|---|---|---|
| 2015 | 2015_12 | December 9, 2015 | 70,511,308 |
| 2016 | 2016_11 | November 30, 2016 | 92,558,090 |
| 2017 | 2017_12 | December 20, 2017 | 128,263,573 |
| 2018 | 2018_11 | December 5, 2018 | 168,593,206 |
| 2019 | 2019_11 | December 11, 2019 | 213,522,593 |
| 2020 | 2020_06 | December 2, 2020 | 261,174,669 |
| 2021 | 2021_04 | November 17, 2021 | 280,483,851 |
| 2022 | 2022_05 | December 14, 2022 | 323,519,324 |
| 2023 | 2023_05 | November 8, 2023 | 376,564,447 |
| 2024 | 2024_06 | November 27, 2024 | 435,574,000 |

Table 4: The selected UniRef100 releases in our benchmark. The number of proteins listed are the numbers listed on the UniRef website, before we do any processing and deduplicating.

### F.3 DATA FILTERING EXPERIMENTS WITH LARGER MODELS

In Figure 20, we conduct a similar data filtering experiment as in Figure 3a, but with a larger model (350M parameters) in order to verify if the results hold at a larger scale or if they were an artifact of the smaller model potentially saturating performance. We select the intersection that had the best performance in the smaller model experiments (2014 intersected with 2016), and compare it to training on only 2014 data and only 2016 data. With the small model, we saw that training on the intersection outperformed training on either year alone. In Figure 20, we see the same trend, indicating that this is not an artifact of model size, and rather it is likely about the data quality itself.

## G HYPERPARAMETER SEARCH GRID

In Table 3, we describe the hyperparameter ranges and the selected value for each hyperparameter that was searched over in our study. Each hyperparameter was sampled independently, and we evaluated 16 trials for each method.

## H UNIREF STATISTICS

In Table 4, we list the specific releases we used to construct our benchmark.

