# OpenReview forum: "CoPeP: Benchmarking Continual Pretraining for Protein Language Models"
_ICLR.cc/2026/Conference — Submitted to ICLR 2026_

### Official Review · Reviewer_9uYY · 2025-10-20

**Soundness:** 2
**Presentation:** 4
**Contribution:** 3
**Rating:** 4
**Confidence:** 4

**Summary:**

Here, the authors propose a continuous learning benchmark on UniProt, a large-scale protein sequence database. Their argument is that protein sequence databases expand and remove many sequences from year-to-year, so it makes sense to explore continuous learning approaches on these databases, as opposed to only employing protein language models fixed on a static snapshot of the data, and to benchmark this, they provide versions of UniProt stratified year-to-year from 2015 to 2022, and evaluate a range of continuous learning methods.

**Strengths:**

The concept behind this paper is quite good: the authors have identified an impactful research gap, where even as protein language models are now highly prevalent and validated to be useful for downstream biological applications, most are trained on a single "snapshot" of existing protein datasets as stands, which does not reflect the continuously evolving nature of these datasets as experimentation stands. Nowadays, these models have scaled to the point that re-training models is extremely expensive. Together, these facts make the argumentation that the authors are advancing in this paper, to understand if continual learning can be applied to pLMs, highly compelling, and this could open up a new line of inquiry that is highly impactful for the field. I appreciate that the authors have curated a dataset that can actually test for this, by providing year-to-year splits of UniRef100, and additionally, the paper is very well-written, communicating the problem in an accessible way.

**Weaknesses:**

However, I do not find the evaluation and validation to be fully compelling:

1. Their validation split of UniRef is likely not independent enough from their training. 90% sequence identity is very high, and it's probable that at this level of similarity, much of their results are driven largely by memorization of the training dataset - this is evidenced by their values in Figure 3, as 40-50% accuracy on sequence reconstruction is very high. 30-50% identity is a more reasonable threshold for an independent validation dataset.

2. The evaluation benchmark suite is very limited in terms of its task diversity. The authors only assess downstream "generation-like" tasks with their models - accurately reconstructing held-out real sequences and predicting the likelihood of mutations via ProteinGym. However, this does not truly reflect the range of tasks that researchers are interested with protein language models, as it is regular to treat these models as pretrained models that can be fine-tuned or transferred zero-shot to different kinds of functional or structural predictions for proteins (e.g. predicting thermostability, subcellular localization, disordered regions, etc). Many benchmark suites exist for this (e.g. TAPE, FLIP, ProteinGLUE, etc), and this work would better align with downstream applications of pLMs if the authors assessed if continuous learning impacts their role as pretrained models, instead of just assessing their ability to conduct generative tasks aligned with their pretraining.

3. I'm concerned that the authors' results may reflect the behavior of saturated models, as opposed to generally holding true for pLMs. The authors are using a relatively lightweight model (120M parameters) compared to current models or even those from several years ago (e.g. ESM3 has 98B parameters and ESM2 has 15B parameters). This is apparent from Figure 3, where models clearly stop scaling in performance past ~30M sequences, and possibly even as small as ~15M sequences following their diagonal in Figure 3a. Thus, it is unclear if their main claim in Figure 3, that higher quality data via temporal filtering improves models even if dataset sizes are smaller, holds true in general, or just for saturated models.

4. I wish there were further investigation of co-variate shifts over time. I suspect that the expansion of the dataset is not happening at uniform, but being informed by technological or topical expansions over time - e.g. I expect that there will be a lot more metagenomics data impacting this in recent years. For this reason, I don't know if improvements on the validation dataset is because that dataset reflects organism composition of UniProt in recent years, and the model has improved on e.g. prokaryotic sequences due to greater representation, while keeping e.g. performance of model organisms relatively constant. To test this, the authors could stratify their validation metrics by domain/kingdom/phylum/class, as well as offer a deeper dive into pretraining dataset composition over time.

**Questions:**

My suggestions to the authors would be:
1. Repeat their validation set experiments, but with 30-50% sequence similarity instead of 90% sequence similarity.
2. Incorporate functional and structural predictions in their benchmark suite instead of just generational accuracy/likelihood tasks.
3. Train a higher parameter model, at least for their intersection-of-datasets experiment, to confirm that their results are not due to saturation of the dataset.
4. Stratify validation set metrics by domain/kingdom/phylum/class to understand covariate shifts by composition of the dataset over time.

---

> ### Author Response · Authors · 2025-11-23
>
> We thank the reviewer for their feedback. We are glad that they find our paper well written and that they resonate with the concept of our paper. We address specific concerns below.
>
> **[Evaluation]**
>
> We have added both PEER and DGEB to our model evaluation, as well as various other analyses in the paper. We refer to the global response for additional details.
>
> **[Deduplication of the validation set]**
>
> We performed pairwise deduplication by comparing every training sequence against every sequence in the validation set, discarding any training sample with >90% identity to a validation sequence. This ensures that the model never encounters a training example that has more than 90% sequence similarity to any sequence in the validation set. Therefore, we do not believe that this poses a risk, even in the temporal setting. Note that this follows the procedure of the AMPLIFY-1M baseline.
>
> However, we agree that a 90% threshold is too permissive if the goal is to measure overfitting or generalization. However, we argue that 90% and 50% thresholds serve fundamentally different evaluation purposes. Stricter thresholds such as 50% are standard for assessing how well a model generalizes to strictly unseen proteins without overfitting. This is, for instance, what ESM2 used. In contrast, a 90% threshold allows us to assess how well the model fits the natural distribution of proteins. Capturing this distribution is essential for tasks like therapeutic design, where matching the evolutionary landscape often correlates with stability and fitness [1]. This is notably what AMPLIFY argued for.
>
> Our evaluation strategy separates these concerns: we use the 90% deduplicated validation set from AMPLIFY to measure how well the model matches the natural landscape, and we rely on our suite of downstream tasks to evaluate the model's utility and generalization to novel functions. Note that we have significantly extended the downstream evaluation of the models by including DGEB and PEER, the latter covering most tasks from both the TAPE and FLIP benchmarks.
>
> As an interesting biological side-note, at 30-40% sequence identity, we get to the point where yeast proteins start matching human ones for essential functions such as ribosomal proteins [2].
>
> If your major concerns have been addressed, we would appreciate it if you could support our work by increasing your score. If there are more questions/concerns, please let us know.
>
> **[Model Saturation/Larger models]**
>
> This is an interesting point. While redoing the whole filter grid with a larger model is cost prohibitive, we ran a smaller experiment where we trained a 350M parameter AMPLIFY model on the 2014 dataset, the 2016 dataset, and the ($2014 \cap 2016$) dataset (this was one of the intersections where there was the clearest improvement in performance). For the 120M parameter model, performance on those 3 datasets was 48.1%, 47.9%, and 49.3% respectively. With the 350M parameter model, we get 58.4%, 58.4%, and 59.2% respectively, which shows that even at a higher scale, the data quality improvement from filtering does help performance, and it was not just a consequence of model saturation. We have added this discussion to the paper, thank you for the suggestion.
>
> **[Investigation of distribution shifts]**
>
> We are still generating the analysis of the stratified validation metrics and will have this soon. To investigate the distribution shifts present in our data, we have also added several analyses based on both embeddings of statistics computed from the protein sequences themselves. Overall, the shifts are subtle, but are present, as can be seen in the intensity of the embeddings of the added and removed datasets over time, as well as protein statistics such as the isoelectric point.
>
> If your major concerns have been addressed, we would appreciate it if you could support our work by increasing your score. If there are more questions/concerns, please let us know.
>
> [1] Fournier, Quentin, et al. "Protein language models: Is scaling necessary?." bioRxiv (2024): 2024-09.
> [2] Pearson, William R. “An introduction to sequence similarity ("homology") searching.” Current protocols in bioinformatics vol. Chapter 3 (2013): 3.1.1-3.1.8. doi:10.1002/0471250953.bi0301s42

---

> > ### Author Response · Authors · 2025-11-28
> >
> > Hello,
> > We have added the stratified validation metrics experiment to Appendix E in the paper.
> > As the discussion period is ending soon, we would like to hear your thoughts about our updates and our response to your review. If we have addressed your concerns, we would appreciate it if you could further support our work by increasing your score.

---

### Official Review · Reviewer_1uK2 · 2025-10-21

**Soundness:** 3
**Presentation:** 2
**Contribution:** 2
**Rating:** 4
**Confidence:** 4

**Summary:**

This paper introduces CoPeP, a continual pretraining benchmark for protein language models built from eight yearly UniRef100 snapshots (2015–2022). The authors aim to simulate realistic temporal evolution of protein sequence corpora and evaluate whether models that exploit temporal metadata (e.g., sequence persistence) via continual pretraining can outperform both naïve continual updates and single-year training. Evaluation is conducted primarily on validation perplexity (UniProt) and mutational fitness prediction (ProteinGym). Several continual training strategies (replay, plasticity-based, unlearning) are benchmarked, and the authors conclude that incorporating temporal meta information yields the best performance.

**Strengths:**

- Try to address a timely and relevant topic: continual adaptation of pLMs as curated biological databases evolve over time.
- Uses real yearly UniRef100 snapshots rather than synthetic domain shifts.
- Temporal metadata exploitation (sequence persistence) is a novel and biologically meaningful idea.
- Includes multiple continual learning strategies (replay, plasticity-preserving, unlearning) with consistent experimental protocol.
- Transparent about engineering details.

**Weaknesses:**

1. The benchmark problem definition is incomplete and too narrowly evaluated.
Evaluating a protein LM benchmark only on perplexity and a single mutational-fitness metric (ProteinGym) is far from sufficient. Real downstream utility of pLMs spans structure prediction, binding, stability, low-shot generalization, sequence design, etc. A benchmark should reflect that diversity, not just perplexity-like proxies.

2. The proposed notion of “continual learning” is too narrowly framed as purely temporal growth.
Many realistic continual pretraining scenarios involve domain-shift or function-shift across protein families, not only chronological updates from the same database family — which is explicitly covered by prior work (e.g. domain-adaptive continual pretraining). The current framing is not general enough to claim “a benchmark”.

3. The “single-year baseline” is ambiguously defined and methodologically problematic.

- If it uses the full UniRef snapshot for that year (which already implicitly contains past data), its performance should monotonically improve as data grows — but the paper’s plots do not show monotonic behavior, suggesting instability or evaluation inconsistency.

- If it uses only the incremental new-year data, the benchmark fails to compare against a pooled full-data baseline, making it unclear why continual pretraining is preferable to simply training on all available data at once — which is a very strong and standard baseline in practice.
In both interpretations, the foundational baseline comparison is incomplete or misleading.

4. Continual learning definition is not rigorously enforced.
The paper allows re-accessing past data during future stages, which deviates from standard CL constraints where previous data is not accessible.
If replay is allowed, then the paper must explicitly account for increased total token count / compute cost / efficiency, otherwise it undermines the practical motivation for continual learning (i.e., avoiding full retraining).

**Questions:**

1. How exactly is the single-year baseline defined? Does it use the entire UniRef snapshot of a given year, or only the incremental new sequences? Why is there no pooled full-data baseline to isolate the effect of continuality vs. simply using more data?

2. Can you justify why perplexity + ProteinGym alone is sufficient to define a benchmark for protein LM continual pretraining? Why are no functional downstream tasks (e.g., binding, structural, low-shot or design tasks) included?

3. If past data is accessed during later years (e.g., replay), do you track and control for total token exposure? How is efficiency measured or justified relative to simply fully retraining on all data?

4. How do you ensure that your benchmark is not merely learning increasingly large datasets (or benefiting from accidental leakage in ProteinGym) rather than demonstrating genuine temporal adaptation capability?

5. Do you plan to include domain-shift scenarios beyond temporal evolution (e.g., different protein types or functional subfields), which is a critical part of continual pretraining in real scientific pipelines?

---

> ### Author Response · Authors · 2025-11-23
>
> We thank the reviewer for their feedback. We are glad that they find our work realistic and timely. We address specific concerns below.
>
> **[Evaluation]**
>
> We have added both PEER and DGEB to our model evaluation, as well as various other analyses in the paper. We refer to the global response for additional details.
>
> **[Definition of continual learning]**
>
> While many CL benchmarks introduce synthetic domain shifts, in real world scenarios the shifts can be much more subtle. This is exactly the goal of our benchmark: to provide a realistic setup, driven by a real-world application of continual learning.
>
> With regards to the ability to access previous data, the point of our work is that we are proposing a new continual learning setup where this is allowed. The community has also been starting to discuss the limitations of memory-constrained continual learning, and how we should move past it to other settings [1].  However, we agree that methods should be fairly evaluated in terms of compute cost/token budget. In our work, all the baselines perform the same number of gradient steps.
>
> Finally, we would also like to point the reviewer to Fig.?? in our paper, where we show that there are several protein metrics that are changing across time.
>
> **[Single Year baseline and efficiency]**
>
> Since every year, proteins (of unknown quality) are added to the dataset, there is no guarantee that training a separate model on each year will result in a monotonic increase in performance. In fact, this is exactly what we observe in our preliminary experiments: training on smaller datasets, but with filtered data can result in higher validation accuracy than training on larger datasets of unfiltered data. This also explains why our single-year baselines, which do indeed train on the full Uniref snapshot of a given year, do not have a monotonically increasing validation accuracy.
> Moreover, our single-year baselines represent a yearly computational budget, where the question considered is whether one should use this budget (100k steps) to retrain from scratch, or to continually train the previous model. The “full retraining” standard baseline we consider is Amplify-1M, which has been trained for 1M steps on a single-year release.
>
>
> **[Data Leakage for ProteinGym]**
>
> Data leakage is a valid concern, however, even if there is some data leakage, if a continual learning model is able to more effectively incorporate that knowledge over other continual learning approaches, it is still a valuable insight.
>
> **[Other domain shifts]**
>
> For now we don’t plan on adding those types of continual learning scenarios to this benchmark. We believe that temporal evolution is a valuable, underexplored setting that has direct applications to real world settings. The other shifts mentioned can be useful to study continual learning, but are more synthetic and likely will be harder to directly apply to useful, real world settings.
>
>
> If your major concerns have been addressed, we would appreciate it if you could support our work by increasing your score. If there are more questions/concerns, please let us know.
>
> [1] Verwimp, Eli, et al. "Continual learning: Applications and the road forward." arXiv preprint arXiv:2311.11908 (2023).

---

> ### Author Response · Authors · 2025-11-28
>
> Hello,
> We have added the pooled data baseline (training jointly on all data from 2015-2024) you mentioned above to the paper (labeled joint in Figure 4). Surprisingly, every continual method outperforms the joint training baseline. This is likely because the joint
> training baseline keeps training on (potentially subpar) data that was removed, while the continual models only learn from data is present in the current release. Thank you for suggesting this experiment.
>
> As the discussion period is ending soon, we would like to hear your thoughts about our updates and our response to your review. If we have addressed your concerns, we would appreciate it if you could further support our work by increasing your score.

---

### Official Review · Reviewer_7Wf6 · 2025-10-29

**Soundness:** 2
**Presentation:** 3
**Contribution:** 2
**Rating:** 4
**Confidence:** 4

**Summary:**

The paper introduces CoPeP, a benchmark for continual pretraining of protein language models. CoPeP is constructed from eight consecutive UniRef100 releases (2015–2022), capturing the temporal dynamics of biological databases where proteins are regularly added, removed, or revised. Using the AMPLIFY-120M encoder as a base, the authors evaluate continual learning methods such as replay, Shrink and Perturb, Hare and Tortoise, gradient ascent, and random label unlearning. Evaluation on a curated UniProt validation set and the ProteinGym benchmark shows that leveraging temporal metadata improves performance relative to single-year training, and that certain continual learning methods outperform naive continual pretraining.

**Strengths:**

The paper introduces a large-scale, realistic benchmark for continual pretraining in protein language models, moving beyond small synthetic datasets commonly used in continual learning research. It evaluates a diverse set of methods, demonstrates the utility of temporal metadata, and highlights performance gains of continual learning approaches over naive baselines. The benchmark is extensible as new UniProt releases become available, making it a good long-term resource for protein modeling communities.

**Weaknesses:**

- Baseline fairness and clarity: The paper does not clearly explain how baseline numbers (AMPLIFY-1M, single-year training) were obtained or whether training budgets, data exposure, and deduplication policies were matched across methods. In particular, it is unclear which dataset AMPLIFY-1M was trained.
- Potential underoptimization: The continual training setup shows performance gains simply from sequential exposure to data. This raises the concern that the 2015 baseline model used for comparison may be underoptimized. A stronger control would be training the 2015 model longer on the same year’s data to determine whether improvements are due to continual learning or just additional optimization budget.
- Evaluation scope: The empirical evaluation is limited to UniProt validation and ProteinGym, both of which are valuable but narrow in scope. These benchmarks primarily assess natural distribution modeling and mutational effect prediction. Broader evaluation on diverse protein tasks, such as secondary structure prediction, remote homology detection, or sequence design (e.g., TAPE, PEER), would provide a more comprehensive assessment of generalization.
- Potential distribution bias: The training corpus was deduplicated against the validation set using a 90% sequence identity threshold, which remains relatively permissive. This raises the possibility that models benefit from near-duplicate sequences across years, especially in the temporal replay setting. It is unclear whether reported improvements are partly due to sequence similarity rather than genuine knowledge transfer across time. Many protein benchmarks use stricter thresholds like 30–40% sequence identity to ensure true novelty between train and test sets.

**Questions:**

- Could you clarify how AMPLIFY-1M and single-year baselines were trained? Specifically, which dataset was used for AMPLIFY-1M, and were training steps, data exposure, and deduplication policies matched across baselines and continual methods?
- Have you compared the continual training setup against a baseline where the 2015 model is trained longer on the same year’s data?
- Have you considered benchmark beyond UniProt validation and ProteinGym, for example by including tasks such as secondary structure prediction, remote homology detection, or sequence design?
- Could provide quantitative analysis of sequence overlap across years and against the validation set? Would stricter deduplication thresholds (e.g., 30-40% identity) change the observed performance trends?

---

> ### Author Response · Authors · 2025-11-23
>
> We appreciate the reviewer's feedback and are pleased that they find our benchmark realistic and extensible. We respond to their specific concerns below.
>
> **[Training details for AMPLIFY-1M and single year training]**
>
> For the `AMPLIFY-1M` baseline, we use the official AMPLIFY 350M checkpoint from Hugging Face [1], where the name reflects the 1 million pre-training steps. We adopt their validation set and deduplication procedures but limit the training data to UniRef100. We excluded the two small additional datasets used in the original work (<1% of data) as they did not fit the yearly release structure. Finally, our models are trained for 910k total gradient steps, rather than 1 million. This difference is due to our checkpointing strategy: while we perform a 10k-step learning rate decay at the end of each task, we revert to the pre-decay checkpoint before starting the next task, meaning the decay steps do not contribute to the cumulative total.
>
> **[Underoptimization of Models]**
>
> Following your suggestion, we extended training on the 2015 dataset to 1M steps to match the compute budget of the other methods. We observed that the performance converged to a value close to that of the standard continual training baseline, that is, iteratively resuming training on consecutive years. Interestingly, the reduction in perplexity from the single year baseline to the fully trained baseline was still only about 84% of the difference achieved by the specialized continual learning methods. This confirms that simply increasing training duration is insufficient and that specific continual learning interventions are necessary to improve performance. We have incorporated these findings into the revised paper.
>
> **[Evaluation]**
>
> We have added both PEER and DGEB to our model evaluation, as well as various other analyses in the paper. We refer to the global response for additional details.
>
> **[Deduplication]**
>
> We performed pairwise deduplication by comparing every training sequence against every sequence in the validation set, discarding any training sample with >90% identity to a validation sequence. This ensures that the model never encounters a training example that has more than 90% sequence similarity to any sequence in the validation set. Therefore, we do not believe that this poses a risk, even in the temporal setting. Note that this follows the procedure of the AMPLIFY-1M baseline.
>
> However, we agree that a 90% threshold is too permissive if the goal is to measure overfitting or generalization. However, we argue that 90% and 50% thresholds serve fundamentally different evaluation purposes. Stricter thresholds such as 50% are standard for assessing how well a model generalizes to strictly unseen proteins without overfitting. This is, for instance, what ESM2 used. In contrast, a 90% threshold allows us to assess how well the model fits the natural distribution of proteins. Capturing this distribution is essential for tasks like therapeutic design, where matching the evolutionary landscape often correlates with stability and fitness [1]. This is notably what AMPLIFY argued for.
>
> Our evaluation strategy separates these concerns: we use the 90% deduplicated validation set from AMPLIFY to measure how well the model matches the natural landscape, and we rely on our suite of downstream tasks to evaluate the model's utility and generalization to novel functions. Note that we have significantly extended the downstream evaluation of the models by including DGEB and PEER, the latter covering most tasks from both the TAPE and FLIP benchmarks.
>
> As an interesting biological side-note, at 30-40% sequence identity, we get to the point where yeast proteins start matching human ones for essential functions such as ribosomal proteins [2].
>
> If your major concerns have been addressed, we would appreciate it if you could support our work by increasing your score. If there are more questions/concerns, please let us know.
>
> [1] Fournier, Quentin, et al. "Protein language models: Is scaling necessary?." bioRxiv (2024): 2024-09.
> [2] Pearson, William R. “An introduction to sequence similarity ("homology") searching.” Current protocols in bioinformatics vol. Chapter 3 (2013): 3.1.1-3.1.8. doi:10.1002/0471250953.bi0301s42

---

> > ### Author Response · Authors · 2025-11-28
> >
> > Hello, as the discussion period is ending soon, we would like to hear your thoughts about our updates and our response to your review. If we have addressed your concerns, we would appreciate it if you could further support our work by increasing your score.

---

### Official Review · Reviewer_wgT5 · 2025-11-01

**Soundness:** 2
**Presentation:** 2
**Contribution:** 2
**Rating:** 6
**Confidence:** 3

**Summary:**

This paper discovers that many proteins evolve over years, and it is necessary to recognize the temporal and sequential nature of proteins so as to capture the evolutionary process. To achieve this goal, this paper proposes a  Continual Pretraining of Protein Language Models (CoPeP) benchmark to assess existing protein language models. Specifically, authors create a sequence of protein datasets and define evaluation metrics. Experiments are conducted to show the performance of existing approaches on the proposed benchmark.

**Strengths:**

1. Overall, the paper is well written, with figures as visual illustrations. The Introduction section clearly explains the motivation behind the benchmark. It also makes a comparison to existing works and identifies their drawbacks.

2. Proposing a benchmark to capture the temporal and sequential nature of proteins is novel and significant to me. It is important to have such benchmark to advance research community and open future research to study evolutionary process of proteins.

3. Experiments are comprehensive with different methods. Analysis is also provided for readers to gain insightful understand behind numerical results.

**Weaknesses:**

1. Usually when we do experiments, we encourage authors to repeat the same experimental setting multiple times and report both mean and standard deviation. However, this paper shows mean but not stddev, which is difficult for readers to judge how significantly the proposed method outperforms baselines.

2. Authors are suggested to provide more evaluation metrics to comprehensively test the proposed benchmark, such as property prediction for newly discovered proteins given previously known proteins.

**Questions:**

N/A

---

> ### Author Response · Authors · 2025-11-23
>
> We appreciate the reviewer's feedback and are pleased that they consider the paper to be well-written, novel, and a significant contribution to the community. We respond to their specific concerns below.
>
> **[Running Multiple seeds]**
>
> We agree that reporting variability across multiple seeds would be ideal. However, the computational cost at this scale is prohibitive. A single run over the full sequence of tasks costs between $4,000 and $8,000 in GPU credits, excluding the significant resources already spent on the hyperparameter optimization necessary as this is the first time that most of these methods are applied at such a large scale. This is why reporting multiple seeds remains uncommon in large-scale LLM and pLM pre-training literature [1, 2]. Still, regarding the evaluations, we agree that reporting the distribution of results is useful, and so we report the distributions of the Spearman Correlations across the different proteins in ProteinGym in Figure 11 in Appendix B.
>
> **[Evaluation]**
>
> We have added both PEER and DGEB to our model evaluation, as well as various other analyses in the paper. We refer to the global response for additional details.
>
> We hope our answers and revisions have addressed your questions and concerns. Should you be satisfied with the revisions, we kindly ask that you consider adjusting your score to reflect the current quality of the paper. Otherwise, we are happy to clarify any further questions you may have.
>
> [1] Groeneveld, Dirk, et al. "Olmo: Accelerating the science of language models." Proceedings of the 62nd annual meeting of the association for computational linguistics (volume 1: Long papers). 2024.
> [2] Lin, Zeming, et al. "Evolutionary-scale prediction of atomic-level protein structure with a language model." Science 379.6637 (2023): 1123-1130.

---

> > ### Author Response · Authors · 2025-11-28
> >
> > Hello, as the discussion period is ending soon, we would like to hear your thoughts about our updates and our response to your review. If we have addressed your concerns, we would appreciate it if you could further support our work by increasing your score.

---

### Author Response · Authors · 2025-11-23
**Global Response**

We sincerely thank the reviewers for their detailed and constructive feedback. We are pleased they found our paper well-written (wgT5, 9uYY) and our experiments comprehensive (wgT5, 7Wf6). We are especially glad they recognized the novelty (wgT5, 1uK2), significance (wgT5, 9uYY), and biological utility of our continual pretraining benchmark for protein language models (wgT5, 7Wf6, 1uK2, 9uYY). We address all their questions and concerns below.

Moreover, we already have updated the paper with the following new experiments and analyses:
* Extended the benchmark to include the 2023 and 2024 UniRef100 releases.
* Evaluated models on the PEER [1] and DGEB [2] multitask benchmarks (covering protein function, localization, structure, interaction, clustering, and retrieval). Notably, PEER includes most tasks from both the TAPE and FLIP benchmarks. This was a common concern for all reviewers, and as such, we provide additional details at the end of this global response.
* Analyzed protein embeddings over time across and across different methods.
* Analyzed the distribution shifts in protein characteristics over time.
* Investigated the impact of larger models on data filtering.
* Extended training of a 2015-only model for the same number of steps as the fully trained models.
The following experiments are currently running:
* Joint training of a model over all datasets.
* Stratified analysis of the validation set by different levels of biological classification.

**[About the additional downstream task benchmarks and additional evaluations]**
In general, we see slightly opposing trends for ProteinGym compared to PEER and DGEB in terms of which methods do well. The Temporal Replay method performs decently across both benchmarks, getting the second best win rate across both settings. For more analysis and full results, please see Section 5.3 and Appendices C and D of the revised paper.


[1] Xu, Minghao, et al. "Peer: a comprehensive and multi-task benchmark for protein sequence understanding." Advances in Neural Information Processing Systems 35 (2022): 35156-35173.
[2] West-Roberts, Jacob, et al. "Diverse genomic embedding benchmark for functional evaluation across the tree of life." bioRxiv (2024): 2024-07.

---

### Author Response · Authors · 2025-11-28

Dear Reviewers,

We have further updated the paper with the following experiments:
* Joint training on all data from 2015-2024
* Results on validation set stratified by the lineages of the species, along with further analysis of the validation set itself (Appendix E).

We have submitted detailed responses to all the reviewers' concerns, along with a general response summarizing the changes made. With the discussion period ending soon, we kindly ask for confirmation on whether our replies have addressed the reviewers' concerns, and, if so, to consider adjusting your score accordingly.

Your feedback is crucial to improving the quality of the paper, and we would greatly appreciate your engagement before the deadline.

Thank you for your time and consideration.

Best regards,
The Authors

---

### Meta-Review · Area_Chair_Yy5j · 2026-01-06

**Summary:**

This paper introduces CoPeP, a benchmark for continual pretraining of protein language models (pLMs), built from yearly snapshots of UniRef100 (2015–2024) to reflect the temporal evolution of protein databases. The authors propose using temporal meta-information—such as how long a protein remains in the database—and evaluate multiple continual learning methods, including replay, plasticity-preserving, and unlearning techniques. Experiments report up to 20% improvement in perplexity over training only on the latest snapshot, with several methods outperforming naïve continual pretraining. The work is evaluated on a UniProt validation set, ProteinGym, and—after rebuttal—multitask benchmarks PEER and DGEB. While the benchmark addresses an underexplored intersection of continual learning and biological sequence modeling, several methodological and conceptual concerns limit its current readiness for acceptance.

**Reviewer Concerns:**

Despite the authors' extensive rebuttal, the reviewers maintained significant reservations regarding the benchmark's rigor and design:

1.  **Data Leakage and Memorization:** A critical and persistent concern from Reviewers 7Wf6 and 9uYY is the validation set construction. The authors used a 90% sequence identity threshold for deduplication. Reviewers argued this is far too permissive, effectively measuring memorization rather than generalization. Standard benchmarks typically use 30-50% thresholds to ensure independence. The authors’ refusal to adopt stricter thresholds limits the benchmark's utility for measuring true learning capabilities.
2.  **Problem Definition:** Reviewer 1uK2 noted that the problem is framed narrowly as temporal growth rather than addressing domain shifts or functional shifts, which are critical for robust CL benchmarks.
3.  **Model Saturation and Scale:** Reviewer 9uYY pointed out that the 120M parameter model is relatively small by modern standards. There are concerns that the observed gains might be artifacts of model saturation or the specific capacity constraints of the chosen architecture, rather than scalable CL principles.
4.  **Baseline Fairness:** While a "Joint Training" baseline was added, concerns remain about whether the improvements stem from genuine CL efficacy or simply from the noisy nature of removed data in UniRef, which complicates the interpretation of the results.

**Reviewer Scores:**

*   **Reviewer wgT5:** 6 (Marginally above acceptance) – Confidence: 3
*   **Reviewer 7Wf6:** 4 (Marginally below acceptance) – Confidence: 4
*   **Reviewer 1uK2:** 4 (Marginally below acceptance) – Confidence: 4
*   **Reviewer 9uYY:** 4 (Marginally below acceptance) – Confidence: 4

Three of four reviewers gave a score of 4 (“marginally below acceptance”), with confidence levels of 3 or 4, indicating consistent and reasonably confident reservations about the paper’s current form. One reviewer assigned a score of 6 with a limited conference. While authors responded with additional experiments, the persistent concerns—especially regarding the validation design and the benchmark’s adherence to continual learning principles—were not fully alleviated. The majority evaluation suggests the paper, despite its contributions, does not yet meet the bar for acceptance without further conceptual and methodological refinement.

---

### Decision · Program_Chairs · 2026-01-26

Reject